



# An ice sheet model validation framework for the Greenland ice sheet

Stephen F. Price[1], Matthew J. Hoffman[1], Jennifer A. Bonin[2], Ian M. Howat[3], Thomas Neumann[4], Jack Saba[4,6], Irina Tezaur[5], Jeffrey Guerber[4,7], Don P. Chambers[2], Katherine J. Evans[8], Joseph H. Kennedy[8], Jan Lenaerts[9], William H. Lipscomb[1], Mauro Perego[10], Andrew G. Salinger[10], Raymond S. Tuminaro[10], Michiel R. van den Broeke[9], and Sophie M. J. Nowicki[4]

[1]Los Alamos National Laboratory, MS B216, Los Alamos, NM 87545, USA.
[2]University of South Florida, St. Petersburg, FL 33701, USA.
[3]The Ohio State University, Columbus, OH 43210, USA.
[4]NASA Goddard Space Flight Center, Greenbelt, MD 20771, USA.
[5]Sandia National Laboratories, P.O. Box 969, MS 9159, Livermore, CA 94551, USA.
[6]Science, Systems, and Applications, Inc., Lanham, Md 20706, USA.
[7]Sigma Space Corp., Lanham, MD 20706, USA.
[8]Oak Ridge National Laboratory, MS 6301, Oak Ridge, TN 37831, USA.
[9]Utrecht University, Utrecht, Netherlands.
[10]Sandia National Laboratories, P.O. Box 5800, MS 1320, Albuquerque, NM 87185, USA.

*Correspondence to:* Stephen Price (sprice@lanl.gov)

**Abstract.**

We propose a new ice sheet model validation framework – the Cryospheric Model Comparison Tool (CMCT) – that takes advantage of ice sheet altimetry and gravimetry observations collected over the past several decades and is applied here to modeling of the Greenland ice sheet. We use

realistic simulations performed with the Community Ice Sheet Model (CISM) along with two idealized, non-dynamic models to demonstrate the framework and its use. Dynamic simulations with CISM are forced from 1991 to 2013 using combinations of reanalysis-based surface mass balance and observations of outlet glacier flux change. We propose and demonstrate qualitative and quantitative metrics for use in evaluating the different model simulations against the observations. We find

that the altimetry observations used here are largely ambiguous in terms of their ability to distinguish one simulation from another. Based on basin- and whole-ice-sheet scale metrics, the model initial condition as well as output from idealized and dynamic models all provide an equally reasonable representation of the ice sheet surface (mean elevation differences of <1 m). This is likely due to their short period of record, biases inherent to digital elevation models used for model initial conditions, and biases resulting from firn dynamics, which are not explicitly accounted for in the models

or observations. On the other hand, we find that the gravimetry observations used here are able to unambiguously distinguish between simulations of varying complexity, and along with the CMCT, can provide a quantitative score for assessing a particular model and/or simulation. The new frame-





work demonstrates that our proposed metrics can distinguish relatively better from relatively worse

simulations and that dynamic ice sheet models, when appropriately initialized and forced with the right boundary conditions, demonstrate predictive skill with respect to observed dynamic changes occurring on Greenland over the past few decades. An extensible design will allow for continued use of the CMCT as future altimetry, gravimetry, and other remotely sensed data become available for use in ice sheet model validation.

**1   Introduction**

Over the past few decades, high spatial and temporal resolution remote-sensing-based observations have been collected over Earth's large ice sheets. These observations include wide spatial coverage and unprecedented detail concerning, for example, the present state of, and changes in, ice sheet surface velocity (Joughin et al., 2010; Moon et al., 2012), ice sheet surface elevation and rates of

elevation change (Pritchard et al., 2009; Shepherd et al., 2012; Csatho et al., 2014), direct measurements of the rate of ice sheet mass change (e.g., Jacob et al., 2012; Sasgen et al., 2012; Velicogna and Wahr, 2013; Wouters et al., 2013), and have allowed for synthesis assessments of overall ice sheet mass balance (Shepherd et al., 2012). While some of these observations have been used for the validation of ice sheet models – testing whether ice sheet model outputs are consistent with observations

(e.g., Aschwanden et al., 2013, 2016) – the significant knowledge barrier to understanding and using remote sensing data and the lack of a standard framework for comparing available observations with model output makes such comparisons difficult and far from standard practice.

    We present a new ice sheet model validation framework – the Cryospheric Model Comparison Tool (CMCT) – that aims to fill this gap. Broadly, the CMCT software is designed to post-process

model output over a specified time frame, process and filter available spatially and temporally coincident observations from remote-sensing based datasets, compare the two, and assess the model versus observation mismatch using a number of proposed qualitative and quantitative metrics. Here, we demonstrate the CMCT using observations over the Greenland ice sheet obtained between 2003 and 2013 from the Ice, Cloud, and land Elevation Satellite (ICESat) and from the Gravity Recovery

and Climate Experiment (GRACE) satellites. The design is, however, intentionally extensible to allow for the incorporation of other similar observational datasets (e.g., for the Antarctic ice sheet) covering similar or longer time periods (e.g., ice sheet surface altimetry from other space- and airborne missions and follow-on missions to ICESat and GRACE). Implicit in our development of the CMCT is the understanding that the observational datasets of interest may be very large, may entail

complex processing for which ice sheet modelers have little or no expertise, and may be updated, appended to, or altered in numerous ways at any point in the future. As such, data are accessed remotely via an online interface (https://ggsghpcc.sgt-inc.com/cmct/) that insures a CMCT user is able to take advantage of data processing improvements and new data sets as they become available.



While we acknowledge that validation should be "model agnostic" – that is, the framework makes
no assumptions about the type of mesh used by the model (structured versus unstructured) – the
prototype discussed below assumes regular gridded output. Planned improvements will eventually
allow for model output on unstructured meshes.

The paper proceeds as follows. We first describe the ICESat and GRACE observations used for
model validation, as well as additional datasets that we use here for model forcing, or to provide
constraints on dynamic ice sheet model simulations. We then briefly describe the models, idealized
and dynamic, that we use to generate outputs for comparison to observations. A detailed description
of our dynamic modeling approach is then presented, including our model initialization procedure
and the way in which we force the model to produce realistic output for comparison to observations.
Next, we discuss the processing steps undertaken by the CMCT, so that both the observational data
and model simulation outputs can be compared. We then discuss and present examples for the
qualitative and quantitative validation "metrics", which are provided as outputs from the CMCT.
By using the CMCT to evaluate our model outputs, we demonstrate that our proposed metrics are
successful at distinguishing relative levels of skill exhibited by different simulations. We finish with
discussion and conclusions.

## 2    Observational Data

We use a number of different observational datasets, for model validation and for initializing and
forcing our model simulations. These data are introduced briefly here and, where relevant, discussed
in additional detail below.

### 2.1    Model Validation: ICESat Observations

For model validation, we use time series of ice sheet surface elevation based on data from the Geo-
science Laser Altimeter System (GLAS) aboard the ICESat satellite. GLAS was the first (and is so
far the only) space-borne laser altimeter used for Earth observations (Zwally et al., 2002). Instru-
ment problems forced a campaign-based rather than continuous observational mode within months
of its launch in early 2003. From 2003 to 2006, there were three 1-month campaigns per year, in
February to March, May to June, and October to November. For 2007-2009, this was reduced to two
campaigns per year, skipping the May to June campaign.

Here, we started with the GLA12 standard product files, release 634 (Zwally et al., 2014). From
these we extracted the elevation, reflectivity, waveform fit uncertainty, and instrument gain for par-
ticular locations. The data were filtered as described below and all points (i.e., individual data points
from GLAS) were required to:

- have valid latitude, longitude, and elevation

- lie within the ice sheet area, based on the GIMP 90-m ice mask for Greenland (Howat et al.,



2014) or the ASAID coastline (including ice shelves) for Antarctica (Bindschadler et al., 2011)

- have a waveform fit with one and only one peak

- have elevation within 200m of the elevation at the nearest GIMP 90-m Digital Elevation Model (DEM) node for Greenland (Howat et al., 2014) or Bamber 1-km DEM node for Antarctica (Bamber et al., 2009) (to eliminate outliers due to clouds and blowing snow)

- have reflectivity and waveform fit standard deviation limited by campaign-dependent values (using the same limits as in the first IMBIE analysis (Shepherd et al., 2012))

The DEM elevations (GIMP or Bamber) are also stored, as are the "cleaned" data, for use in model-to-observation comparisons. Additional parameters included in the edited datasets include the surface slope, an estimate of the elevation uncertainty, a drainage system identifier, elevations relative to the WGS84 and EGM08 datums, and for Antarctica, a flag indicating whether a measurement is on the continent or an ice shelf. All relevant data are archived in NetCDF format and are self-
documenting.

Here, for comparison to the model outputs discussed below, we picked three "snapshots" of Greenland ice sheet surface elevation, with dates of 2003.8, 2004.8, and 2007.8. For model output, the decimal year corresponds to a time slice for which model output was written. For observations, it corresponds to observations averaged over the month of October. The years 2003, 2004, and 2007
were chosen because they allow for the largest number, the widest spatial coverage, and the highest quality of surface elevation retrievals, and being near the end of the melt season, also minimize elevation biases due to snow cover and firn.

### 2.2    Model Validation: GRACE Observations

For model validation, we also use time series of ice sheet mass change, as recorded by the GRACE
satellites. GRACE, launched in 2002 and still running as of early 2016, is a dual-satellite mission with the primary objective of measuring the gravity field of the Earth as it changes over space and time (Wahr et al., 1998). It does this by very accurately measuring the distance between the two satellites (to within a few micrometers), which are flying one behind the other in the same orbit, about 200 km apart and 400-500 km above Earth's surface. Precisely measuring how the Earth's gravity
tugs each of the satellites differently allows for the calculation of changes in Earth's gravitational potential field. Over time, this allows for estimation of the mass change at repeatedly surveyed locations on the surface.

GRACE has several important limitations. First, only changes in mass can be estimated, not the absolute mass. Thus, GRACE results are always presented relative to something (usually either
the time series starting point or the mean over some time span). Second, while the observed mass changes are due to the sum of solid Earth motions, hydrology changes, ice loss, and atmospheric





effects, GRACE data provide no information about the relative contribution of each term. Here, we assume that solid Earth, ocean, and hydrological changes are small near Greenland, compared to the ice mass change term. Pre-processing of GRACE data has already removed some fraction of the total signal using an atmospheric and a non-tidal ocean model (Flechtner, 2007; Dobslaw et al., 2013).

A final limitation is that GRACE provides data with relatively low spatial resolution. The regularly released, degree-and-order 60 spherical harmonics provide approximately 300 km of resolution (in terms of the half-wavelength) at the equator, and perhaps twice that resolution near Greenland. But that is only if one ignores measurement errors, which grow as spatial resolution gets finer. In practice, some sort of smoothing is typically used, which reduces the resolution to regions of ∼500-1000 km in diameter. Thus, the major difficulty when comparing GRACE to ice sheet model output is scaling the two data sets to comparable spatial resolutions. Here, we address this through "smoothing" of high-resolution model output to GRACE-like resolution, by first converting them to degree-and-order 60 spherical harmonics, and then comparing the observations and smoothed model output directly. Details on the spherical harmonics conversion can be found in Wahr et al. (1998).

The following standard processing steps (Chambers and Bonin, 2012) have been taken with the GRACE observations used herein: geocenter terms added to the $C_{20}$ term are replaced with the higher-accuracy SLR version, and a model is used to remove the signal due to glacial isostatic adjustment (GIA). The monthly data are averaged into yearly increments, from 2003-2012, to match with annually-averaged simulation data. Here, averaging to annual time scales reduces the post-processing burden for both model output and observations, and also eliminates large month-to-month noise in the GRACE signal (so that no additional smoothing or de-striping is required). In principle, however, higher time-resolution observations could be used for model validation.

### 2.3 Model Initialization and Forcing

The initial ice sheet geometry (ice sheet thickness and bedrock topography) is prescribed from Morlighem et al. (2014). Other relevant initial and boundary condition data, used in our model spin-up, include maps of the present day surface temperature and geothermal flux, which are taken from the SeaRISE project (Bindschadler et al., 2013; Nowicki et al., 2013). Surface velocity data, used for optimizing our initial model velocity field, are from Joughin et al. (2010). All data are interpolated onto a uniform, 1 km resolution mesh.

In prognostic runs, we force our model with Regional Atmospheric Climate Model version 2 (RACMO2) monthly surface mass balance (SMB) data from van Angelen et al. (2013). Those data, provided at 11 km resolution, are spatially interpolated onto our 1 km model grid. For one simulation, we also force our model with a time series of outlet glacier flux, which is based on observations and described in more detail in Enderlin et al. (2014).



## 3 Models and Simulations

We use the Community Ice Sheet Model (CISM) version 2.0 (Price et al., 2014; Lipscomb et al., 2016) coupled to *Albany-FELIX*, a three-dimensional, finite-element code for solving the first-order
accurate Stokes approximation (Blatter, 1995; Pattyn, 2003; Dukowicz et al., 2010). Additional details on *Albany-FELIX*, including model verification, can be found in Tezaur et al. (2015a) and Tezaur et al. (2015b). Because the momentum balance solver is called from CISM, the model uses CISM-native routines for evolving the ice sheet geometry and internal temperatures.

In addition to simulations conducted with our dynamic ice sheet model, we construct two ide-
alized and highly-simplified model simulations – one that assumes a geometry fixed in time at the model initial condition and another that evolves only according to SMB forcing. The intent of these idealized models is to provide a baseline for use in quantifying if and how a dynamic ice sheet model presents added value in terms of improving the match to observations. We discuss these idealized models in more detail below.

We demonstrate validation with the CMCT by running these models forward in time during a period coincident with the ICESat and GRACE observations. The models are forced with a combination of observationally-based SMB and/or outlet glacier flux time series.

### 3.1 Model Initial Conditions

We generate initial conditions for our dynamic ice flow model closely following methods discussed
in Price et al. (2011), Nowicki et al. (2013), and Edwards et al. (2014). Briefly, with no-slip basal boundary conditions, we spin-up the internal temperature and velocity fields using the present-day geometry and thermal boundary conditions (noted above). Temperature and velocity are allowed to freely evolve via their coupling through the flow-law rate factor, but we maintain a fixed geometry. After 10,000 yrs, we use this initial temperature field and observed surface velocities (Joughin et al.,
2010), to perform a formal, observation constrained inversion for the basal traction coefficient field in a linear-friction sliding law (as described in more detail in Perego et al., 2014). This updated set of model velocities and basal boundary conditions are then used for additional spin-up of the temperature field, again maintaining a fixed ice sheet geometry. Periods of temperature spin-up are alternated with updates to the basal traction coefficient field and this process continues iteratively
until internal temperatures approach an approximate steady-state (in this case, model temperatures were spun-up for a total of 350 kyrs). The final internal temperature field is then held steady for the decadal-scale simulations conducted here. Observed surface speeds, those from our optimized initial condition, and differences, are shown in Figure 1. While the model fit to observed velocities is reasonably good over the ice sheet as a whole, we note that the model clearly underestimates
speeds for some of the major outlet glaciers (see also Figure 2). These differences are relevant to the discussion below, since these same outlet glaciers encompass the areas experiencing the largest





dynamic changes during the past few decades.

Lastly, we note that while we assign a date of 1991 to our initial ice sheet state, the data used
to define the initial geometry include ice thickness observations spanning the early 1990's to late
2000's (Morlighem et al., 2014) and surface elevations nominally dated to ∼2000 and ∼2007, for
the interior and margins, respectively (Howat et al., 2014) (Further, these surface elevations are based
in part on ICESat observations, as discussed further below in Section 6). Similarly, the velocity ob-
servations we optimize our model to were largely collected during the early- to late-2000's (Joughin
et al., 2010). While these represent the best available datasets, the variable time span associated with
the observations does introduce biases into our results that are difficult to quantify.

### 3.2 Model Simulations

Starting from the initial condition described above, we run our models forward in time from 1991-
2013. For our first simulation, which we refer to as the "SMB-only" simulation, we force the model
with RACMO2 SMB from 1991-2013, in the form of anomalies:

$$SMB(t)_{anomaly} = SMB(t)_{RACMO2} + FC. \tag{1}$$

Anomaly forcing is used here because, following the temperature spin-up and optimization of sliding
velocities, the model is not in equilibrium with the present-day ice sheet geometry and SMB forcing.
Thus, we apply a flux-correction, $FC$, to the modeled SMB fields in order to maintain an initial
steady-state relative to the long-term-average SMB from 1960-1990 (the time series of spatially
integrated, net SMB, relative to the 1960-1990 mean, is implied by the black-dotted line in Figure
4).

The flux correction, $FC$, is calculated as

$$FC = SMB_{SS} - SMB_{mean}, \tag{2}$$

where $SMB_{mean}$ is the 1960-1990 mean SMB, $SMB_{SS}$ is the steady-state SMB field, and the
latter is calculated as the negative of the modeled thickness field change when the model is run
forward a single time step with no SMB forcing applied. SMB is applied to the model in units of
ice-equivalent per year (m yr$^{-1}$), assuming an ice density of 910 kg m$^3$.

In our second simulation, we apply RACMO2 SMB forcing as noted above, but in addition we also
force the model using the Enderlin et al. (2014) time series of observed outlet glacier flux change for
22 of Greenland's largest outlet glaciers, which are responsible for ∼80% of the observed dynamic
ice loss since 2000. Under the assumption that flux changes are dominated by velocity changes
(discussed further below), we apply these changes as anomalies relative to our modeled glacier
velocities by first converting the data to a velocity increase *relative to* the 1999 observations (all 22
outlet glaciers would have a relative flux increase of 1 in 1999, i.e. no increase). We then apply these
relative flux increases as a spatially variable, multiplicative constant directly to our modeled outlet



glacier velocities at the same locations; starting in 1999, we "play back" the converted time series of Enderlin et al. (2014) against our modeled 1999 velocity field and apply them as Dirichlet boundary conditions on the model velocity several km upstream from observed grounding line locations.

We refer to this as the "SMB+FF" (FF="flux forcing") simulation. In some cases, flux gate loca-
tions in our model domain are extended from the grounding line to the ice margin. In all cases, ice downstream of flux gates is removed from comparison with observations in a post-processing step (discussed further below). The locations of flux gates where Dirichlet boundary conditions are applied in our model are shown in Figure 1. The observed versus modeled flux for the 22 outlet glaciers and the time series of relative outlet glacier speedup are shown in Figures 2 and 3, respectively.

Inherent in our treatment of the Enderlin et al. (2014) data is the assumption that the ice flux change over any time step is well approximated by,

$$\Delta(U_i H)\Delta t = (U_i \Delta H + H \Delta U_i)\Delta t \approx H \Delta U_i \Delta t, \tag{3}$$

where $U_i$ is the depth-averaged ice velocity (a vector field in plan view), $H$ is the ice thickness, $t$ is time, and $\Delta$'s imply a finite-difference like approximation. In summary, we assume that the change
in flux is dominated by the change in velocity, and that the advection of changes in ice thickness is a small contribution to the overall flux change at a point. We justify this approximation by noting that, over the time period of interest, variations in speed of these outlet glaciers range from ∼10-100%, whereas fractional changes in ice thickness are an order of magnitude smaller.

All simulations are run on a uniform, 1 km resolution mesh. To maintain stability and accu-
racy when using our explicit, forward-Euler time stepping scheme, we use a time step of 0.025 yr. Monthly SMB anomalies are held steady in time during sub-monthly model time steps. Because both our SMB time series and our optimized basal boundary conditions are limited in coverage to the footprint of the ice sheet in its initial configuration (i.e., that of the present-day) we allow the ice sheet margin to retreat but not advance; at each time step, we apply a mask to simply remove any ice
that has advanced beyond the initial footprint. Both observations and our model experiments support the fact that, over the simulation time period explored here, the dominant mode of ice sheet evolution is strong marginal thinning. Experiments without this treatment lead to problematic ice growth and advance in limited marginal regions and, in reality, strong negative mass balance at and beyond the current ice sheet margin provides a significant barrier to margin advance. As discussed below,
the comparison between model output and observations is only conducted for grid cells within this masked region.

In addition to the two simulations run using our dynamic ice sheet model, we include results from two idealized, non-dynamic models. The first, which we refer to as the "persistence" model, is simply the present-day ice sheet geometry, unchanging in time. The second, which we refer to
as the "RACMO2-SMB-only" model, starts from a present-day ice sheet geometry and, at monthly time steps, RACMO2 SMB fields are applied. Simultaneously a proxy for a steady component of discharge is included by subtracting the 1960-1990 mean SMB field at every time step. As mentioned



above, the purpose of these idealized models is to provide a benchmark against which to compare results from dynamic model simulations.

### 4 Processing of Model Output and Observations

Model output is initially written in NetCDF format (Unidata, 2015). Simple scripts based on the NetCDF Operators toolbox (NCO; see Zender, 2008) are used to extract only the necessary time series of fields from these output files and apply additional post processing operations. We first extract the basal topography (here, time invariant) and the ice thickness fields, and then sum the two to derive a time series of the ice sheet surface elevation. Because the profiles where we apply Dirichlet boundary conditions in our SMB-FF experiment are (in some cases) several to tens of km inland from the present-day ice sheet margin, we apply a post-processing mask to remove any ice thickness downstream of these regions (which otherwise undergo anomalous evolution). In order to then not compare ice thickness change over two different ice sheet domains in the SMB-only versus SMB-FF experiments, we apply this same mask to the time series of thickness data from the SMB-only experiment. The result of this post-processing step is that ∼0.2% of the initial ice sheet domain is removed when we compare model output to observations. To further facilitate comparison to ICESat observations, we extract model output that coincides with the month of October in select years (as discussed above in Section 2.1). For comparison to GRACE observations, we take annual averages of the ice thickness time series (which are recorded at approximately monthly time resolution, see, e.g. Figure 4). These trimmed NetCDF files, containing latitude, longitude, thickness, and thickness or surface elevation, were sent to NASA's Goddard Space Flight Center (GSFC) for processing by the CMCT (As noted above, as of mid-2016, the CMCT is online and files can be uploaded via the internet). The CMCT then performs the comparisons described below.

Since the locations of ICESat data rarely coincide with the model grid points, the four model elevations surrounding each ICESat measurement were interpolated bilinearly in the polar-stereographic space of the model domain to predict the model elevation at the ICESat measurement location. ICESat measurements without four surrounding model nodes were excluded from the comparison. Our goal was to remain as close to the elevation observations as possible, and only apply interpolation at the last possible step.

In contrast, to compare model outputs of ice sheet mass change to those from the GRACE satellites, the majority of the post-processing is applied to model output data. Our aim is to mimic as closely as possible the simulation data, as GRACE would see such a signal from space. This ultimately means filtering model output to arrive at a more limited "GRACE-like" spatial resolution, on a 0.5×0.5° global grid.

To perform this filtering, we first relocate each 1×1 km model grid cell into broader 0.5×0.5° grid cells. If more than one simulated data point is located within a 0.5×0.5° cell, those values are



averaged. Model grid cells near the coast require special care, since the higher-resolution simulated data often does not fully fill each $0.5 \times 0.5\,^\circ$ cell. To avoid over-estimating the coastal signal, we

multiply the averaged signal for coastal cells by the ratio of the total model grid area in that cell to the total cell area (with the latter value fixed at $0.5 \times 0.5\,^\circ$). We then convert ice thickness to the GRACE-like units of equivalent water height, by multiplying by the density ratio of ice to water (910/1000). Simulated grid cells along the coasts that contain no-data values during any year are removed from the computation for all years. The time-mean for each $0.5 \times 0.5\,^\circ$ grid cell is calculated and removed,

since GRACE sees only mass anomalies. This results in data on the same grid as GRACE, and in the same units, but still with far greater spatial resolution than GRACE. When comparing to mass change over a specific region, we also convert this to total mass change in gigatons by taking into account the known area of the region.

   To further reduce the resolution of the processed model output so that it more closely mimics

GRACE observations, we convert from a $0.5 \times 0.5\,^\circ$ uniform grid to spherical harmonics (Wahr et al., 1998). By using spherical harmonics of infinite degree and order, we would perfectly reproduce the simulated output (i.e., the output would look identical to what it looks like on the raw, $0.5 \times 0.5\,^\circ$ grid). However, we instead cut off the spherical harmonics at degree and order 60, which limits the resolution to precisely what can fit into the solution space that GRACE measures. To make later

comparisons easy, we then reconvert the limited spherical harmonics back to a spatial map on a $0.5 \times 0.5\,^\circ$ grid.

   We note that in the present work, the GRACE processing and metric calculations (discussed next) were conducted using software developed at the University of South Florida (J. Bonnin and D. Chambers, personal communication). The publicly available service will include this software and

functionality as a processing option when using the CMCT.

## 5   Qualitative and Quantitative Metrics

In this section, we propose metrics for use in characterizing output from a particular simulation relative to the observations. Our qualitative metrics consist of a range of visual outputs (figures) and our quantitative metrics consist of several whole-ice-sheet (and in some cases, basin-specific) scalar

values used to assign an overall score to a particular simulation. Both are made available by the CMCT. In Section 6, we provide these metrics for the current set of simulations and discuss in more detail where they are and are not able to distinguish between relatively better and relatively worse simulations.

### 5.1   ICESat Metrics

For the qualitative comparison to ICESat observations, we present three types of qualitative metrics: (1) map-view plots of the differences in modeled and observed ice sheet surface elevations for any





given year (e.g., Figure 5), (2) histograms of these differences (e.g., Figure 6), and (3) scatter-plots of these differences as a function of the distance from ice sheet grid cell to the nearest ICESat observation point (e.g., Figure 7). All of these figures can be used to quickly identify and fix gross

problems that might occur during a simulation or post processing. As an example, a difference in model and ICESat vertical datums was easily discovered through examination of Figures 5 and 6. Outliers as a result of incorrect masking of model and observational data were easily discovered and corrected by inspection of Figures 6 and 7. Large regional biases would also be obvious from Figure 5 and biases as a result of sparse ICESat sampling would be identifiable using Figure 7. Figure

5 also provides a clear indication of where altimetry data are available or absent for use in model validation.

A quantitative comparison between model output and ICESat observations is given by standard statistics calculated from the distributions shown in Figures 5, 6, and 7. We use the following scalar metrics for the quantitative assessment of a particular year of a particular simulation: (1) the mean

of the elevation differences, $\overline{\Delta z}$, (2) the standard deviation of the elevation differences, $\sigma_{\Delta z}$, (3) the mean of the absolute value of the elevation differences, $\overline{|\Delta z|}$, and (4) the standard deviation of the absolute value of the elevation differences, $\sigma_{|\Delta z|}$ (e.g., Tables 1 and 2 below). These same metrics can be applied to individual drainage basins, as shown in Figure 8.

Examples of these figures and metrics, based on the present work, are shown and discussed in

more detail in Section 6 below.

### 5.2 GRACE Metrics

For the qualitative comparison to GRACE observations, we also present three types of figures: (1) spatial maps showing the observed and modeled ice sheet mass trend over the 2003-2012 time period (e.g., Figure 9), (2) whole-ice-sheet, spatially-averaged plots of mass change, showing the observed

and modeled mass trends as a function of time (e.g., Figure 11, and (3) spatial maps showing the percent of GRACE variance explained by each model simulation (e.g., Figure 12). The first and third maintain spatial information about mass trends at the expense of a collapsed time dimension, while the second maintains temporal information about mass trends at the expense of collapsed spatial dimensions.

The first set of spatial maps displays the total mass change over each $0.5 \times 0.5\,^\circ$ bin near Greenland, for GRACE and as simulated by the various models (Figure 9). These maps have units of meters of equivalent water height lost or gained over the ten years, which differs from ice height depending on the density of snow and firn in a particular region.

The second plot allows us to examine the whole-ice-sheet-averaged mass time series as observed

by GRACE versus as simulated by the models. Because GRACE data is of limited spatial resolution, signals from coastal Greenland tend to smear out into the ocean and into the interior (e.g., Figure 9). If we used a simple mask of Greenland land area to compute an average, we would thus understate



the true mass change. Instead, we create a "ones-and-zeroes" filtering kernel for Greenland (Figure
10, left panel) and transform it to GRACE-like spherical harmonics. The result is a smoothed mask
which is "smeared" in the same way that GRACE observations are (Figure 10, right panel). We use
this smoothed mask to weight the spatial average mass change at each time step when calculating
the time series of average mass change shown in Figure 11.

The second set of spatial maps displays the percent of GRACE's observed variance explained by
each model using a combination of statistics. First, the standard deviation of the GRACE signal,
$\sigma_{GRACE}$, is computed in each $0.5 \times 0.5^\circ$ bin (Figure 12, upper left). Then, the standard deviation of
the difference between GRACE and each model is computed, $\sigma_{[GRACE-model]}$. The percentage of
GRACE's variance explained by the model, $PVE$, is thus given by

$$PVE = \frac{\sigma_{GRACE} - \sigma_{[GRACE-model]}}{\sigma_{GRACE}} \times 100. \tag{4}$$

When calculating $PVE$ values, we create a mask of areas near Greenland (but excluding neigh-
boring islands) where the observed GRACE signal has a standard deviation of at least 15 cm of
water (this leaves only the colored regions in the model-observation comparison panels of Fig-
ure 12 (upper-right and lower panels)). This allows us to include both the signal over Greenland
and the part of the signal that is smeared into the ocean by GRACE's low spatial resolution. The
mask also prevents divide-by-zero errors in Equation 4. A model which perfectly reproduces the
GRACE observations has a $PVE$=100 (the model explains 100% of the observed variance). For
$0 < PVE < 100$, the model simulation correctly captures some of GRACE's observed signal, but
has imperfect amplitude or timing. Isolated regions where the difference between the observed and
modeled variance is *larger* than the observed variance can also exhibit negative $PVE$ values.

Based on these three qualitative assessments, we propose two single-value metrics for use in
quantifying the match between model output and GRACE observations. First, from Figure 11 we
compute $M_{Trend}$, the difference between the spatially-averaged linear trend of GRACE and each
model. Second, we create an ice-sheet-wide average of the $PVE$ statistic for each model-to-GRACE
comparison, $M_{PVE}$. For a simulation that provides a perfect match to the observations, $M_{Trend}$
and $M_{PVE}$ would have values of 0 and 100, respectively. These metrics are listed in Table 3 and
discussed in more detail in Section 6 below.

## 6 Results and Discussion

Figure 4 provides a broad summary of the results for the different model simulations conducted here.
All simulations other than the persistence simulation show a clear seasonal cycle in mass balance as
a function of the SMB forcing and, over inter-annual and longer timescales, all simulations steadily
lose mass with a marked increase in the rate of mass loss starting around or after the year 2000. For
the simulation that includes forcing of outlet glacier flux, an additional increase in the rate of mass
loss can also be observed starting around 2005.



From Figures 5–7, it is clear that the modeled ice surface is biased slightly low relative to observations and this remains true regardless of which year we pick to compare against. Yet in general, aside from a few isolated regions near the margin, the model versus observed surface elevation differences are surprisingly small, with mean differences over the entire ice sheet of $< 1.0$ m in all cases (Tables 1 and 2). The persistence model simulation was specifically designed to address the concern that such a seemingly good match is simply a reflection of an initial condition, which is already a good match to observations, that is then simply carried forward in time. Tables 1 and 2 include whole-ice-sheet metrics based on the ICESat observations and confirm that the persistence model has the best overall statistics, followed by the RACMO2-SMB-only model, with the two dynamic model simulations giving the worst overall statistics. Apparently, at the whole-ice-sheet scale, our proposed ICESat metrics are unable to confirm that a dynamic ice sheet model provides any "value added" relative to a static initial condition or a very simple model driven by SMB considerations alone.

To examine this further, we show additional values of our ICESat $\overline{\Delta z}$ metric in Figure 8, but obtained for specific drainage basins. In this case, the assessment of which simulation scores "best" is much more ambiguous; at the scale of specific drainage basins and depending on the year, either idealized or dynamic-model-based simulations perform better, and no clear pattern emerges. At best, any of the models proposed here does a reasonably good job of mimicking the ice sheet state for the particular dates of ICESat observational data used here. But clearly a strong argument cannot be made for the relative ranking of either the dynamic or idealized models when assessed using these data and metrics.

These ambiguous results are likely explained by a number of factors. First, over the timescales considered here, the changes in the observed and modeled ice sheet surface as a function of the different simulations conducted may be small enough that they cannot be distinguished from one another at the drainage-basin or whole-ice-sheet scale. Second, because of the relative short timespan of ICESat data, we only conduct this comparison at three points in time, separated by a maximum of 5 years. By including other data sets prior to (e.g., ERS-1/2 radar altimetry) and after (e.g., Operation Ice Bridge; ICESat-2) the ICESat period, elevation data will likely be more successful at distinguishing relatively better or worse model simulations. Third, our initial condition is biased in the sense that the DEM it is based on (Morlighem et al., 2014) includes some of the same ICESat observations that we compare our model outputs with (see Howat et al., 2014). This likely explains the good match between the ICESat observations and the persistence model, and in this sense, it is encouraging that the much more realistic, but also much more complicated, dynamic models obviously do no harm to that initially good match (Figure 8). Lastly, we make no accounting here for the complications that might be introduced by snow and firn dynamics. However, we note that the model initial conditions are based on observational data that include snow and firn, as do the ICESat data that are used in the comparison. Seasonally, the elevation does vary by a few tens of centimeters



in the middle of the ice sheet and up to a few meters near the ice sheet margins (Kuipers Munneke et al., 2015). This seasonal amplitude is only partly accounted for when converting SMB into the ice-equivalent units used as model forcing.

The comparison to surface elevation snapshots from ICESat provides both whole-ice-sheet and regional information about how well our model simulations mimic the ice sheet *state* at a particular

moment in time. Complementary information regarding how well our simulations mimic the observed *trends* (in this case, mass trends) is provided by the comparison with GRACE observations. Importantly, this comparison also removes any concern about the issue of persistence, since trends for the persistence model are always exactly equal to 0. Spatial maps of the observed and modeled mass trends confirm that all of the simulations capture at least some fraction of the observed mass

loss occurring along Greenland's coasts (Figure 9). No model simulates as large a magnitude of loss along the western or southeastern coasts as GRACE sees and all of the models over-estimate the mass loss in the southwest relative to GRACE. Simulations that include an approximation of ice dynamics by including the time series of outlet glacier forcing show a modest improvement at mimicking GRACE along the southeast, west, and northwest coasts. The dynamic model simulations

have $PVE$ near 50, demonstrating that they see the majority of the GRACE signal in most places (Figure 12). Addition of a dynamic ice sheet model and / or forcing to approximate the effects of outlet glacier dynamics increases the $PVE$ in northwestern, western, and southeastern Greenland, and for the ice sheet as a whole, as shown by the change in the whole-ice-sheet $M_{PVE}$ metric (Table 3). However, dynamic ice sheet models also appear to result in locally negative $PVE$ values in

southwestern Greenland. For the spatially-averaged, whole-ice-sheet mass trend comparison (Figure 11), there is a clear improvement when moving from the RACMO2-SMB-only model, to the SMB-only dynamic model, to the SMB+FF dynamic model, and this same improvement is clearly captured by the ice-sheet-wide $M_{Trend}$ metric (Table 3).

Overall, our qualitative and quantitative GRACE metrics appear to provide a much clearer distinc-

tion between idealized and dynamic model simulations. Further, they also appear to demonstrate an increase in model "skill" in the sense that might be expected; dynamic models perform better than non-dynamic models, and dynamic models that account for known changes in ice dynamics perform better than those that account for only SMB forcing.

## 7 Conclusions

In the present work, we have proposed a software framework – the Cryospheric Model Comparison Tool (CMCT) – for the purposes of ice sheet model validation, which allows us to take advantage of several decades worth of satellite-based observations of the Greenland ice sheet. Based on ICE-Sat altimetry observations and GRACE gravimetry observations, we have proposed qualitative and quantitative metrics for use in assessing ice sheet model skill with respect to mimicking these same



observations. Using both idealized and dynamic models, we have demonstrated the ability of these
metrics to distinguish between relatively better and relatively worse simulations. For the ICESat
observations and simulations conducted here, the related ICESat metrics are unable to unambigu-
ously distinguish between the different simulations. On the other hand, the GRACE observations
and metrics are clearly able to distinguish between relatively worse and relatively better simula-

tions. Importantly, we note that while the comparison with GRACE proves more useful here for
distinguishing model skill with respect to observed ice sheet *trends*, that information would be much
less compelling in the absence of the ICESat comparison, which provide initial confidence that the
models are also skillful with respect to representing the observed ice sheet *state*; different types of
observations have different strengths and multiple sets of observations used in conjunction will allow

for a much more robust validation of models.

All of the simulations conducted here underestimate the observed mass loss from Greenland with
respect to the observations. This occurs for several reasons. First, the match between modeled and
observed velocities is imperfect, and we underestimate ice flux in some important regions in our
initial condition (Figure 2). The result is that modeled changes in ice flux, which are proportional

to the initial flux, are also likely to be underestimated. Since those changes are largely responsible
for the dynamic thinning that has been observed, our model underestimates that thinning as well.
Second, our simulations implicitly assume a steady-state ice sheet in 1991, which is at odds with
recent work suggesting that the ice sheet was already thinning dynamically at that time (Kjeldsen
et al., 2015). Lastly, our forcing of outlet glaciers does not start until 1999, yet we know that dynamic

thinning for several important outlet glaciers (e.g., Jacobshavn, (Joughin et al., 2004) started prior
to that time. In addition to the underestimation of mass loss by the modeling conducted here, it is
worth noting that GRACE estimates of Greenland mass loss are likely to be slight overestimates due
to the same "leakage" effects discussed above. In particular, mass loss from Baffin and Ellesmere
Islands, which are also significant (Gardner et al., 2011), affect GRACE observations of Greenland

mass loss.

The forcing of the dynamic ice sheet model is admittedly overly constrained in the present study
– we have specified both the "passive" climate forcing (in the form of surface mass balance) and the
"dynamic" climate forcing (in the form of outlet glacier flux). In reality, the latter is argued to be
largely a response to the coupling between marine terminating outlets and the surrounding oceans

(see reviews by Straneo et al., 2013; Straneo and Heimbach, 2013), which we ignore here aside
from attempting to include the ice sheet response to that forcing. One can imagine more meaningful
validation exercises in which the ice sheet model is freely evolving and the climate forcing – the ice
sheet surface balance and submarine melting at marine terminating margins – is supplied through
complete coupling with an Earth System Model and translated to a dynamic response through the

appropriate ice sheet model physics (e.g., subglacial hydrological forcing in response to increased
surface melting, iceberg calving in response to submarine melting, etc.).



Nevertheless, for the purposes of demonstrating the effectiveness of the CMCT validation framework at assessing model skill, the present approach has proven effective. Further, the present approach allows us to speculate that, given "perfect" knowledge of climate forcing, coupling, and the
model physics necessary to translate those couplings to the appropriate ice dynamical responses, the dynamic model tested here clearly demonstrates some level of "skill" at reproducing the observations; non-dynamic models perform worse than dynamic models and dynamic models forced only by climate (here, SMB) perform worse than dynamic models forced by both climate and ice dynamics. Put another way, present-day ice sheet models with adequate representations of physics and
boundary conditions, and when forced by realistic climate histories, can be expected to skillfully reproduce observed ice dynamical changes on decadal timescales. This marks a clear improvement over a decade ago, when sea-level rise projections from ice sheet models were not included in the IPCC's 4th Assessment Report (Solomon and others, 2007) because models of that time clearly lacked skill at explaining or mimicking observed ice dynamical behaviors.

Practically speaking, the framework and metrics proposed here would be most useful in a relative sense, for example in quantifying improvements within a specific model (or within a class of models) as a result of differences in model dynamics (e.g., shallow versus higher-order dynamical approximations), model physics (e.g., representations of ice sheet rheological or basal processes), or model resolution (mesh resolution and / or changes in the spatial resolution of input datasets). The
framework and metrics could also be fairly easily adapted for use as a model-to-model intercomparison tool, simply by swapping outputs from another model as the observational data sets. Such development is planned in support of the new Ice Sheet Model Intercomparison Project (ISMIP6; see http://www.climate-cryosphere.org/activities/targeted/ismip6) for CMIP6 (Nowicki et al., 2016 (in prep.).

Lastly, we note that once the model and observational data are both available within the same data structures and co-located on the same grid, any number of additional or alternative metrics can be imagined for comparing and contrasting model output and observations; the main contribution of the CMCT is to make those comparisons possible in the first place.

Future work on the CMCT will focus on the extension to Antarctica, allowing for the use of model
output on unstructured meshes, the addition of other datasets for use in model validation – including new altimetry data from OIB, Cryosat, and ICESat2, older radar-altimetry data from Envisat and the ERS missions – and new gravimetry observations from GRACE2. For comparing observed and modeled surface elevations, improvements could be made by accounting for firn dynamics on both the observation and model sides. On the model side, improvements can be expected to follow from
the use of higher-spatial resolution or downscaled SMB forcing, improved initialization techniques that do not require anomaly SMB forcing (Perego et al., 2014) or that allow models to start from a state that matches observed transients (Goldberg et al., 2015).



## 8 Code and Data Availability

The Community Ice Sheet Model code is available at http://oceans11.lanl.gov/cism/index.html. For
the *Albany* momentum balance solver, please see the code availability statement in Tezaur et al.
(2015a). The raw ICESat and GRACE data discussed above are available for download at
http://nsidc.org/data/icesat/data.html and http://podaac.jpl.nasa.gov/datasetlist?search=GRACE, re-
spectively. The CMCT online service is available at https://ggsghpcc.sgt-inc.com/cmct/ and the
CMCT source code will be made available upon request. Model forcing and initial condition data
sets are available through direct contact with the respective authors listed in the main body of the
text.

*Acknowledgements.* Support for SFP, MJH, IT, KJE, JHK, WHL, MP, AGS, and RST was provided through
the Scientific Discovery through Advanced Computing (SciDAC) program funded by the U.S. Department of
Energy (DOE), Office of Science, Advanced Scientific Computing Research and Biological and Environmental
Research Programs. Support for IMH was provided by NASA Cryospheric Sciences grant NNX11AR47G.
SFP was also partially supported by NASA Cryospheric Sciences and SFP and MJH were partially supported
by the National Science Foundation, under grant ANT-0424589 to the Center for Remote Sensing of Ice Sheets
(CReSIS). JL and MvdB acknowledge funding from the Polar Program of the Netherlands Organization for
Scientific Research (NWO/NPP), the Netherlands Earth System Science Center (NESSC) and Water, Climate,
Ecosystems research theme of Utrecht University. JL is supported by NWO ALW through a Veni postdoctoral
grant. This research used resources of the National Energy Research Scientific Computing Center (NERSC;
supported by the Office of Science of the U.S. Department of Energy under Contract DE-AC02-05CH11231).



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

**Tables and figures**

| Date (yr) | $\overline{\Delta z}$ (m) | $\sigma_{\Delta z}$ (m) | $\overline{|\Delta z|}$ (m) | $\sigma_{|\Delta z|}$ (m) |
|---|---|---|---|---|
| | CISM, RACMO2, Pers. | CISM, RACMO2, Pers. | CISM, RACMO2, Pers. | CISM, RACMO2, Pers. |
| 2003.8 | 0.26, 0.20, 0.15 | 10.48, 10.43, 10.38 | 4.46, 4.40, 4.37 | 9.49, 9.45, 9.42 |
| 2004.8 | 0.11, 0.01, -0.05 | 10.87, 10.83, 10.78 | 4.61, 4.56, 4.52 | 9.84, 9.82, 9.79 |
| 2007.8 | 0.33, 0.19, 0.04 | 10.33, 10.28, 10.24 | 4.35, 4.29, 4.26 | 9.38, 9.34, 9.32 |

**Table 1.** Whole-ice-sheet metrics for model versus ICESat Observations (ICESat - model) for the SMB-only simulation.

| Date (yr) | $\overline{\Delta z}$ (m) | $\sigma_{\Delta z}$ (m) | $\overline{|\Delta z|}$ (m) | $\sigma_{|\Delta z|}$ (m) |
|---|---|---|---|---|
| | CISM, RACMO2, Pers. | CISM, RACMO2, Pers. | CISM, RACMO2, Pers. | CISM, RACMO2, Pers. |
| 2003.8 | 0.27, 0.20, 0.15 | 10.49, 10.43, 10.38 | 4.46, 4.40, 4.37 | 9.50, 9.45, 9.42 |
| 2004.8 | 0.12, 0.01, -0.05 | 10.89, 10.83, 10.78 | 4.62, 4.56, 4.52 | 9.86, 9.82, 9.79 |
| 2007.8 | 0.36, 0.19, 0.19 | 10.34, 10.28, 10.24 | 4.36, 4.29, 4.26 | 9.38, 9.34, 9.32 |

**Table 2.** Whole-ice-sheet metrics for model versus ICESat Observations (ICESat - model) for the SMB+FF simulation.





| Simulation | Trend (Gt yr$^{-1}$) | $M_{Trend}$ (error) | $M_{PVE}$ (%) |
|---|---|---|---|
| GRACE | -186.1 | 0 (0%) | 100 |
| RACMO2-SMB-only | -83.3 | -102.8 (55%) | 39.8 |
| SMB-only | -100.4 | -75.7 (41%) | 46.5 |
| SMB+FF | -121.0 | -65.1 (35%) | 49.7 |

**Table 3.** Observed and modeled mass trend and scalar metrics $M_{Trend}$ and $M_{PVE}$ calculated for the RACMO2-SMB-only, SMB-only, and SMB+FF models from 2003 to 2012.

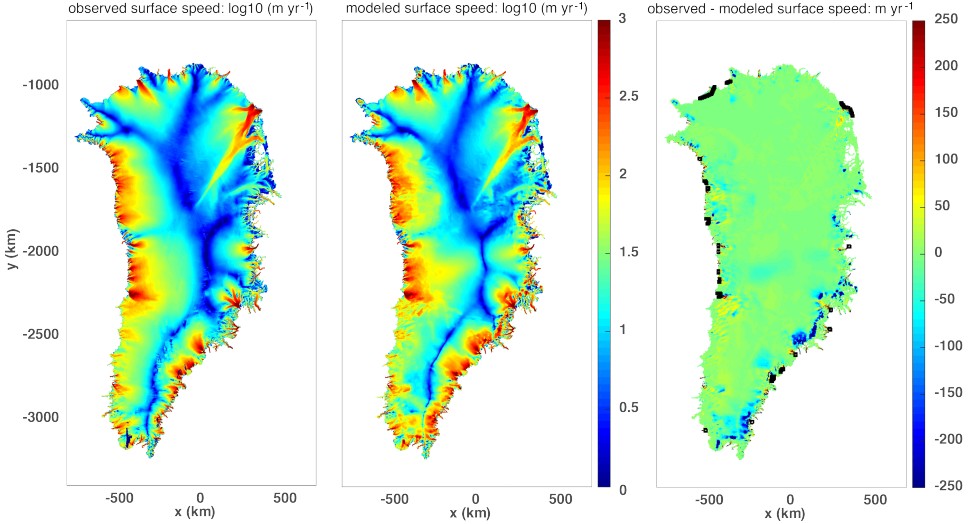

**Fig. 1.** Observed surface speed derived from InSAR observations (Joughin et al., 2010) (left), optimized, modeled surface speed (center), and observed minus modeled surface speed (right). Small black squares in right panel mark the locations of flux gates where Dirichlet boundary conditions on velocity are applied.



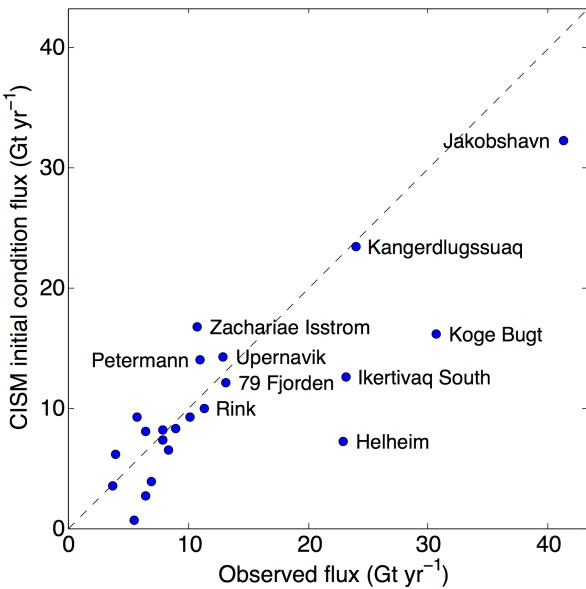

**Fig. 2.** Observed outlet glacier flux from Enderlin et al. (2014) plotted against that from the model initial condition. Outlet glacier locations are marked in the right-hand panel of Figure 1.

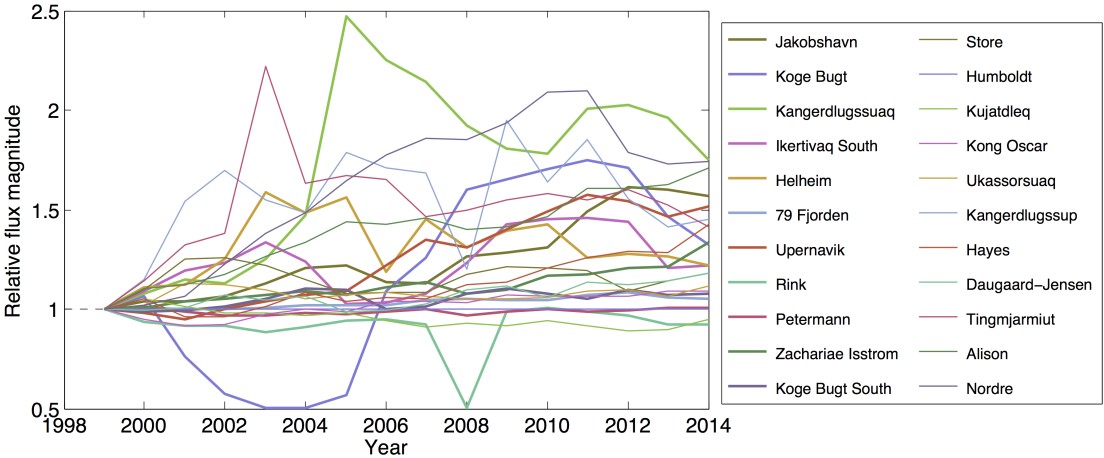

**Fig. 3.** Time series of relative speed-up applied to the modeled outlet glaciers marked in the right-hand panel of Figure 1. Glaciers are ordered in the legend from greatest to least flux in 1999.



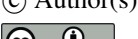

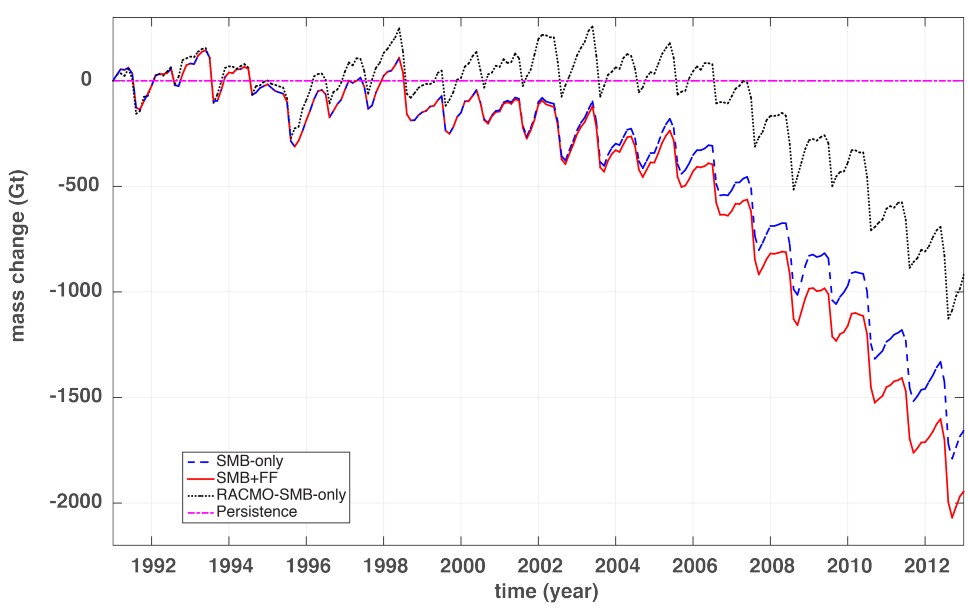

**Fig. 4.** Modeled ice sheet mass change (Gt) versus time relative to the initial condition. Different line types represent different simulations and models, as identified in the legend.



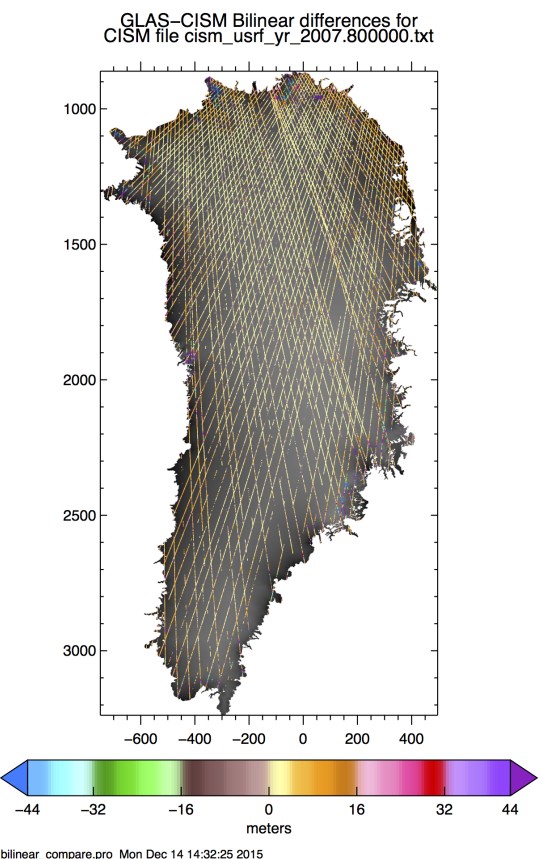

**Fig. 5.** Map of ICESat minus modeled surface elevation differences (m) in 2007.8 for the SMB + FF simu-
lation. Colored lines mark locations of ICESat observations. Grey areas represent regions where no ICESat
observations are available.




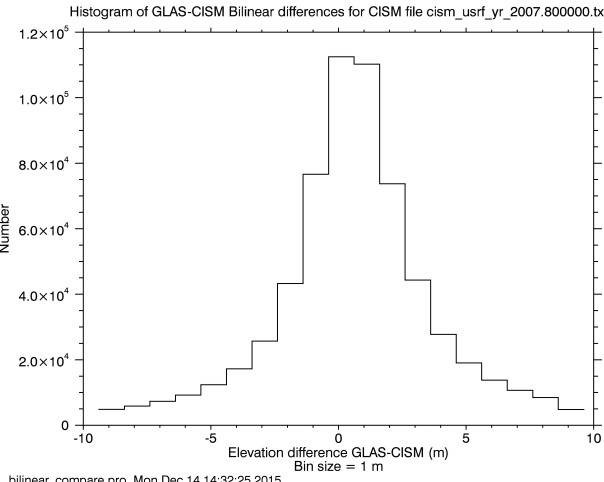

**Fig. 6.** Histogram of ICESat minus modeled surface elevation differences (m) in 2007.8 for the SMB + FF simulation.

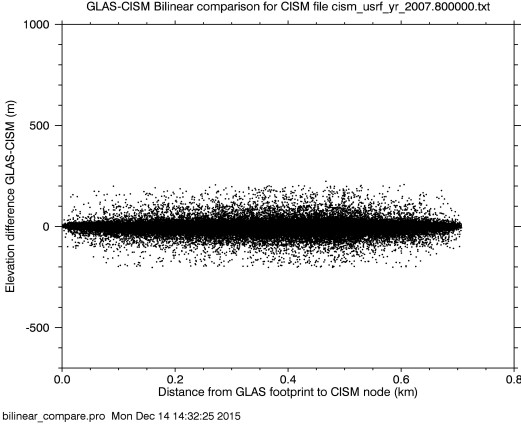

**Fig. 7.** Scatterplot showing difference in model and observation location (km) versus ICESat minus modeled surface elevation differences (m) in 2007.8 for the SMB + FF simulation.





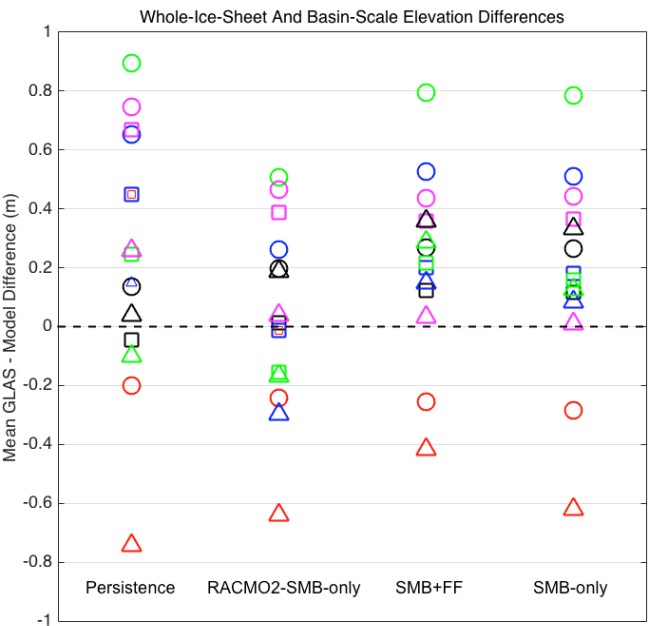

**Fig. 8.** Summary plot for whole-ice-sheet and basin-scale-specific mean elevation differences (GLAS minus model, in meters). Circles, squares, and triangles are for years 2003.8, 2004.8, and 2007.8, respectively. Colors represent different ice sheet areas as follows: entire ice sheet (black), Jacobshavn drainage basin (blue), Kangerlussuauaq drainage basin (red), Helheim drainage basin (green), and the Northwest coast drainage basin (magenta). For 2004.8, all Kangerlussuauaq basin comparisons (missing red squares) plot at around -2.25 m, and are omitted here for clarity of plotting. Drainage basins areas are identical to those defined by IMBIE (Shepherd et al., 2012).



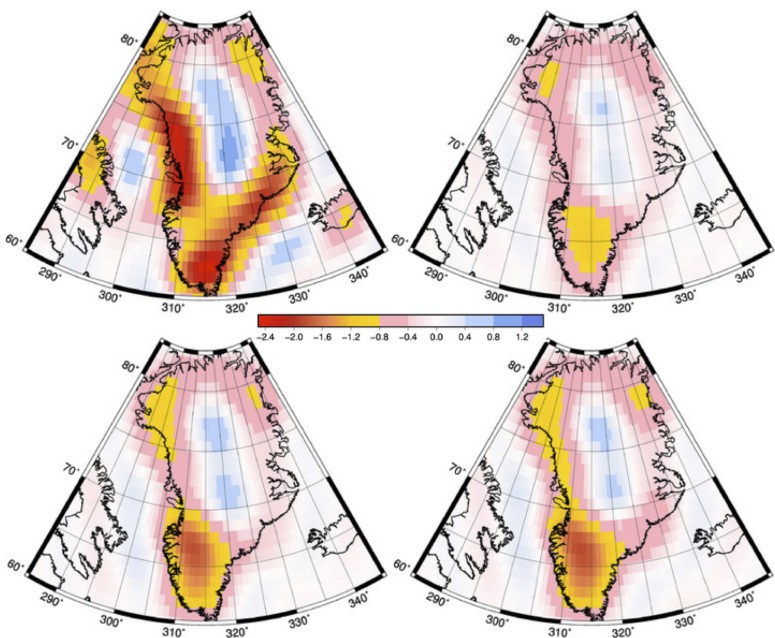

**Fig. 9.** Total mass change between 2003-2012 for GRACE observations (upper-left), RACMO2-SMB-only model (upper-right), SMB-only simulation (lower-left), and SMB+FF simulation (lower-right). Units are meters of water equivalent height.

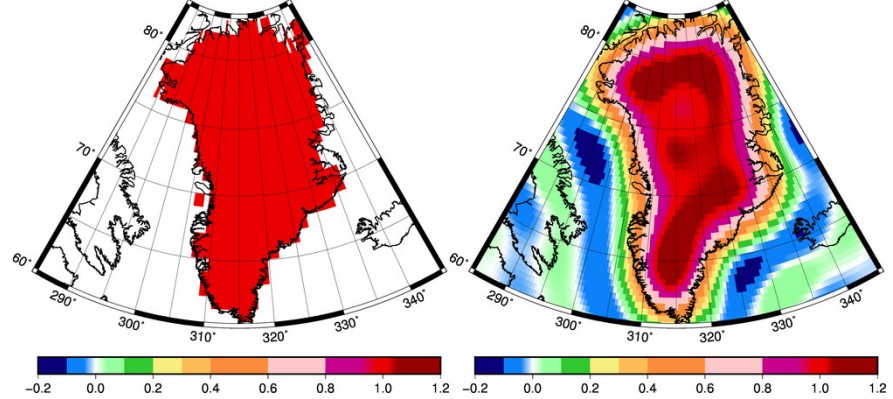

**Fig. 10.** The Greenland kernel (left) and its degree and order 60 smoothed version (right), used when calculating the whole-ice-sheet mass trends shown in Figure 11.




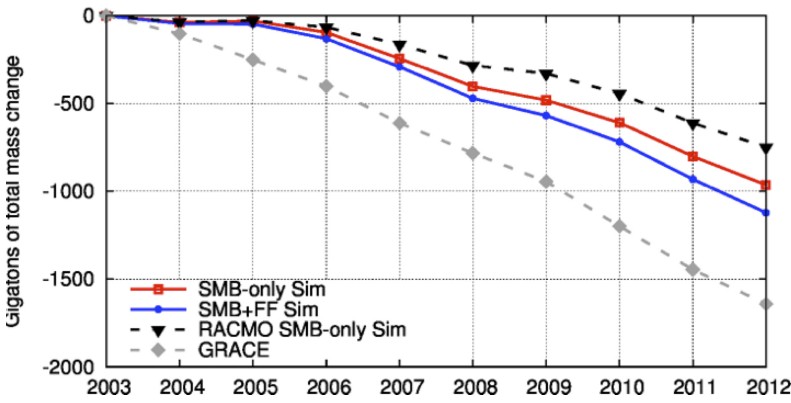

**Fig. 11.** Observed and modeled, whole-ice-sheet mass trends obtained using annually averaged ice thickness fields and the spatial smoothing kernal from Figure 10.

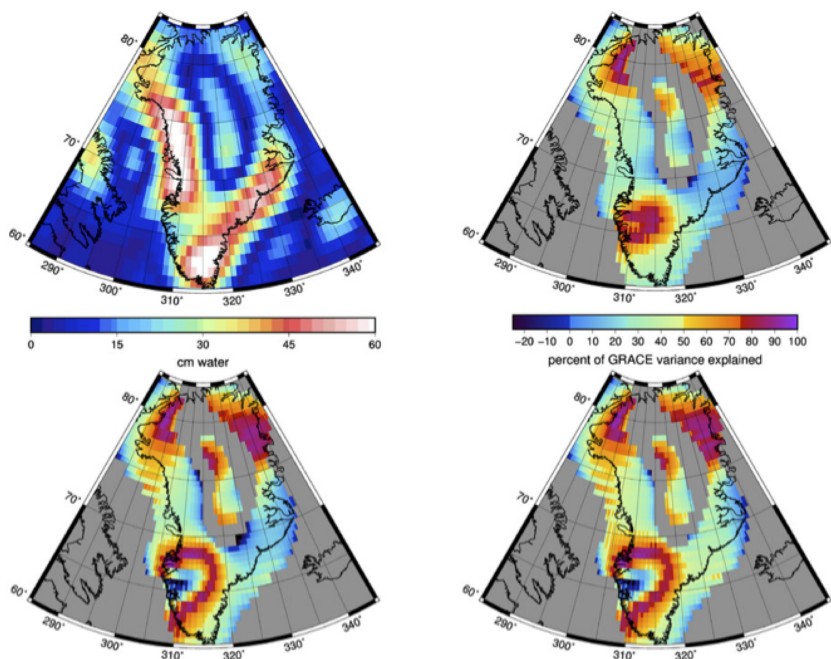

**Fig. 12.** GRACE variance (upper-left) and percent of GRACE variance explained by various models: (upper-right) RACMO2-SMB-only model, (lower-left) SMB-only simulation, (lower-right) SMB+FF simulation, (lower-left). Upper-left colorbar goes with upper-left figure while upper-right colorbar goes with the remaining figures.