# Peer review of "An ice sheet model validation framework for the Greenland ice sheet"

_Geoscientific Model Development, 2016_

## Referee Comment (RC1) · Anonymous Referee #1 · 21 Jun 2016

This paper presents a new tool, the Cryospheric Model Comparison Toll (CMCT), to compare ice-sheet model results to remote sensing observations (altimetry and gravimetry). The paper clearly presents the data sets currently used in CMCT, the processing steps and the maps and metrics produced to validate module outputs. The use of these maps and metrics is illustrated by comparing 4 "model" simulations of the Greenland Ice Sheet between 1991 and 2013. Two of these simulations are non-dynamic, the two others are produce with the Community Ice Sheet Model coupled with Albany-Felix.

This new framework fills a gap between ice-sheet models and remote sensing observations and will allow easier and more efficient model validation and improvement. The paper is well written, with a clear description of the different processing steps and assumptions used for the simulation.

I see maybe one point which could be discussed with more details. For the use and interpretation of the altimetry data, the conclusion is that the proposed metic are not sufficient to discriminate between the simulations. I think this is true on the global scale as surface elevation changes are very small but on very large areas, so GRACE data are more appropriate as they spatially integrate this information. However the altimetry data have a much better spatial resolution that is not used in the metrics proposed here. I think some fine scales metrics could help to discriminate simulations, by example by comparing only points where elevation changes have been significant. Ok clearly 3 data sets over 4 years will not be sufficient but we may expect that things will improve as more data become available.

There is few aditionnal points requiring clarification:

- l210: "(the time series of spatially integrated, net SMB, relative to the 1960-1990 mean, is implied by the black-dotted line in Figure 4).": the time serie in Figure 4 is really the integrated net SMB. I think the "implied" can be removed?

- Bottom of p7, top of page 8: discussion on the dynamic forcing: not sure I fully understand the part "starting in 1999, we "play back" the converted time series...". I understand that there is no forcing from 1991 to 1999, then the forcing is applied as Dirichlet conditions relative to the 1999 model velocities? More precisions are needed for the choice of the location for the Dirichlet conditions. Why "several kilometres upstream from observed grounding line"? How is it chosen? A new figure made from Fig.1 with zooms on particular outlets glaciers could be useful.

- Page 9: "Processing of model Output and observations". I understand that model output should be processed before online submission to the CMCT website. There is no documentation on the website (at least as far as we don't ask for login informations); It would be useful to give a table with the variables, their units, the time dimension, etc ... that should be included in the netcdf for processing by CMCT.

- L355: "e.g. Figure 11", missing closing ")"

- Fig. 10: labels of the colorbars are not visible.

- Fig. 11 caption : "kernal" → kernel

- A paper comparing GRACE data with flow model simulations has just been published in The Cryosphere (Alexander et al., 2016). It could be cited and discussed.

- References are not in alphabetical order; modify with respect to GMD requirements for references.

References:
Alexander, P.M., Tedesco, M., Schlegel, N.-J., Luthcke, S.B., Fettweis, X., Larour, E., 2016. Greenland Ice Sheet seasonal and spatial mass variability from model simulations and GRACE (2003–2012). The Cryosphere 10, 1259–1277. doi:10.5194/tc-10-1259-2016

---

## Referee Comment (RC2) · Anonymous Referee #2 · 22 Jun 2016

The detailed review is in the attached PDF file with the following summary

The manuscript of Price and others describes and applies the method of a validation frameâ–work, called Cryospheric Model Comparison Tool (CMCT), that could potentially be extremely valuable for the validation of contemporary ice sheet model simulations against observations.

The observations comprise currently ICESat ice sheet elevation estimates and GRACE ice sheet mass change estimates. The presented ice sheet simulations of the Greenland ice sheet (GrIS) have used the dynamical "Community Ice Sheet Model" (CISM) versionÂă2 with setups of different complexity and pseudo ice sheet simulations, where the applied spatial distribution of the surface mass balance (SMB) reduce the ice sheet elevation locally. These simulations are compared with the observations to highlight

the added value of using dynamical models beside only applying SMB fields to the elevations and to show how the satellite produces could be used seamlessly to validate ice sheet simulations of the contemporary Greenland ice sheet. This tool box reduces drastically the need for adjusting commonly available satellite products when comparing them with simulations, because it converts the simulations data on the fly to the grids and footprints of the applied satellite products. Detailed ice sheet and basin-wide diagnostics (ice sheet elevation anomalies, mass changes, explained ice mass changes by the simulations), temporal evolutions (elevation differences for few periods and cumulative total mass change), and overall metrics (ice elevation differences and mass trends differences) are provided. The service that is offered as a web service seems to be available after registration; although I have not tried to use this service yet.

The manuscript is very well written, has a clear structure and all tables and figures, which are generally well prepared, are necessary. It was a pleasure to review this manuscript. I hope that the manuscript could be published soon, because I will be extremely helpful to have this information and the offered service.

I recommend the publication of the manuscript after some minor corrections.

Please also note the supplement to this comment:
http://www.geosci-model-dev-discuss.net/gmd-2016-97/gmd-2016-97-RC2-supplement.pdf

**Supplement:**

**Review of Price, Stephan F. et al. (Geoscientific Model Development, gmd-2016-97)**

The manuscript of Price and others describes and applies the method of a validation framework, called Cryospheric Model Comparison Tool (CMCT), that could potentially be extremely valuable for the validation of contemporary ice sheet model simulations against observations.

The observations comprise currently ICESat ice sheet elevation estimates and GRACE ice sheet mass change estimates. The presented ice sheet simulations of the Greenland ice sheet (GrIS) have used the dynamical "Community Ice Sheet Model" (CISM) version 2 with setups of different complexity and pseudo ice sheet simulations, where the applied spatial distribution of the surface mass balance (SMB) reduce the ice sheet elevation locally. These simulations are compared with the observations to highlight the added value of using dynamical models beside only applying SMB fields to the elevations and to show how the satellite produces could be used seamlessly to validate ice sheet simulations of the contemporary Greenland ice sheet. This tool box reduces drastically the need for adjusting commonly available satellite products when comparing them with simulations, because it converts the simulations data on the fly to the grids and footprints of the applied satellite products. Detailed ice sheet and basin-wide diagnostics (ice sheet elevation anomalies, mass changes, explained ice mass changes by the simulations), temporal evolutions (elevation differences for few periods and cumulative total mass change), and overall metrics (ice elevation differences and mass trends differences) are provided. The service that is offered as a web service seems to be available after registration; although I have not tried to use this service yet.

The manuscript is very well written, has a clear structure and all tables and figures, which are generally well prepared, are necessary. It was a pleasure to review this manuscript. I hope that the manuscript could be published soon, because I will be extremely helpful to have this information and the offered service.

**I recommend the publication of the manuscript after some minor corrections.**

**Major Issues**

none

**Minor Issues**

First I give general comments followed by specific comments.

**General Comments**

The currently included and in the future planned observations are based on satellite products, which biases the observation-based validation towards surface properties, such as elevation and its change. What's about ice sheet-internal properties such as temperature profiles from existing ice cores or radar detected layers, that could help to verify the internal structure, which may be important for the susceptibility of the ice sheet to applied forcing in the future?

**Specific comments**

In the following specific comments are made, where "L123" means line 123, for instance.

L52: Could you please add after web-address that the service is available after registration.

L90-21: I'm sorry, but I haven't understood without a doubt what is meant. Please clarify.

L101-107: You may add a quantity such as "number of good data points" to guide the user to identify "good" years as the mentioned years 2003, 2004 and 2007.

L141: Since some may think instead of the calendar year (starting 01.Jan) about the hydrological year (starting 01.Sept). Therefore, could you please clarify.

L151: Interpolation are bi-linear, conservative, …?

L154: Interpolation are bi-linear, conservative, …?

L166: Do you mean:"… and another that evolves *its surface elevation* only according to … ."?

L206: Do you mean:"… and optimization of sliding *parameters to meet observed* velocities, the … ."?

L229-234: Ice beyond the flux gates are not taken into account for the comparison. Does it still exists and interacts with the ice flow upstream?

L243: Do you mean:"… changes in ice thickness are *always* an order of magnitude … ."?

L250: The application of the mask to remove ice beyond the initial footprint acts as a sink. How large is this sink term? How large is its factional size compared to the applied total SMB.

L252-254: You may split the sentence in two pieces:"… advance in limited marginal regions. In reality, strong negative mass balance at … ."

L261: Have I understood you correctly:"… steps, RACMO2 SMB fields are applied *to either increase (positive SMB) or decrease (negative SMB) the ice thickness accordingly.*" ?

L276-277: How large are the 2% compared to the other presented mass changes?

L281-284: Are corner points needed for each grid box. If optional, would the tool box than perform conservative interpolation?

L298: I guess you mean:"… averaged *weighted by grid size.*"?

L299-301: Do you mean:"… cells by the ratio of the *ice-covered* model grid area in that cell to the total cell area… ."?

L302: Is it possible to change the density of the ice on the web page?

L355: You may dropped the closing bracket:"… of time (e.g. Figure 11*)*, and … ."?

L380: You mean probably mean:" … of at least 15 cm of water *over the period 2003-2012* … ."?

L385: You may change:"… GRACE observations has a PVE=100 *at each location* … ."?

L388: Since math (equation 4) is clear you may change:" … the observed variance exhibit negative PVE values"

L391: You may replace $M_{Trend}$ with $\delta M_{Trend}$ to highlight that it is a difference between two trends.

L426: You may write:" … may be *too* small that they … ."?

L436: You may emphasize:"… hat initially good match (Figure 8) *on a period covering a decade.*"

L510: This, of course, only applies if the driving model climate is identical or at least very similar to the current climate. Otherwise I expect that coupled climate-ice sheet models would lead to an ice sheet geometry that differs from the observed state.

L569-694: Reference. What's the style and sorting criteria for Geoscientific Model Development? If none, please sort using the last name and also have a consistent way to write the first names.

**Tables**

Here I refer to the table number

Table 1 and 2: Please give a remark that "Pers" means persistence and you may also indicate "CISM" as Community Ice Sheet Model and RACMO2 as regional model.

**Figures**

The figure number are given.

Figure 1: Could you add to the color bar of the first two left sub-figures (observed and modeled velocities) a second row that lists instead of log10(v) the actual velocity (v)?

Figure 1: In the right figure (velocity difference) the black dots representing the flux gates and the

negative velocity differences (blue color) are hard to distinguish.  What's about adding a tick ("-") slight away from the ice margin to indicate the locations of the flux gates?

Figure 3: To avoid any ambiguity you may change the caption slightly:"… from greatest *(Jakobshavn)* to least *(Nordre)* flux in 1999."

Figure 4: Could you increase the line thickness please?

Figure 7: Since all dots seem to fall within -300 and 300, you may reduce the ordinate axis to this range.

Figure 8: What do you think about increasing the size of the black entire ice sheet symbols?

Figure 9: Since the figure caption indicates a mass change the unit shall be accordingly. You may write:"… Units are meters of water equivalent height / 10 years." or "… Units are meters of water equivalent height for the given period."

Figure 11: You may change the caption:"Observed and modeled, *cumulative* whole-ice sheet trends … "?

Figure 12: To guide the read just spotting the figures what is meant, you may add:" … percent of GRACE variance *(Eqn 4)* explained by … ."?

---

## Referee Comment (RC3) · Anonymous Referee #3 · 28 Jun 2016

**1   General statement**

The manuscript "An ice sheet model validation framework for the Greenland ice sheet" by S. Price and others presents a new framework for comparing ice flow model results with observations of altimetry and gravity acquired since 2003. They apply it to several modeled and prescribed representations of the Greenland ice sheet and show that comparison of results with altimetry observations is similar for all the representations of the Greenland ice sheet, while gravity observations seem to distinguish the more sophisticated representations of the ice sheet evolution.

The paper is clear and well written, the new framework explained in details and the figures usually appropriate. However, there are several major points that are either not

accurate or not supported by the results. My main concern is that the "most sophisticated" dynamic model reproduces the trends in altimetry and gravity for the wrong reasons. Correctly initializing ice flow models remains a challenge today and an active area of research (Seroussi et al., 2011; Aschwanden et al., 2013; 2016) due to the lack of reliable observations and the long response time of ice sheets. In order to bypass this problem, the authors impose an additional steady-state surface mass balance (SMB) on top of the climate derived SMB, therefore forcing the model to remain close to its initial conditions. Without this unphysical forcing, the model would likely diverge quickly from this initial state. Furthermore, the physical model has to be not only forced with an unphysical SMB, but also constrained with imposed velocities over the most dynamic areas, in order to get close enough to the observed changes in gravity. At this point, there is not much physics left in this model.

So if I don't question the development of this framework or the benefits to have a tool that can easily compare observations and model results, I am wondering if any validation of numerical models can be done today given the many improvements still needed by these models to become more accurate. I would therefore recommend to reduce the emphasis of this framework as a validation tool and rewrite the manuscript accordingly.

**2   Specific comments**

In the abstract (l.6) as well as several other places in the manuscript, the author qualify the static representation and the SMB only applied to this static representation as "non-dynamic models", which is misleading, as these representations do not include any physical model. The text should better distinguish between physical models and other representations.

p.1 l.17: "simulations of varying complexity": there is no varying complexity found in the

models presented here (same stress balance approximation, ...). What varies between these simulations is the degree of "forcing" of the models, the velocity being applied as a Dirichlet forcing for one of the simulations.

p.2 l.20-21: The "most sophisticated" models are actually forced by imposing the velocity to be equal to observations at the border of the domain. Showing that such a simulation exhibits mass trends similar to gravity observations does not prove any predictive capability of the model. First, the velocity is not produced by the model but forced in all the dynamic areas. Furthermore, the flux correction applied to the physical models introduces a large mass change that prevents to compare simulations to observations. What this result mainly shows is that observations are rather consistent and that a large part of the mass change signal can be explained by changes in SMB and acceleration.

p.2 l.23: What about velocity changes? I would imagine that comparing the dynamic signal would be an important part of such a tool, as the main objective of an ice sheet model is to reproduce the dynamic signal, not the SMB.

p.2 l.33-37: I don't agree with this statement. The main problem is that ice flow models are still unable to reproduce the observed changes, and that improving their initial conditions is a very active area of research. Models therefore have to either run spin-up or apply flux correction (similar to what is done in this manuscript), to get closer to observations. Accurately comparing model results and observations is therefore beyond the scope of most studies since lots of models cannot even capture the correct trend (Bindschadler et al., 2012).

p.7 l.208: The authors explain how they apply an SMB correction to get closer to a steady-state equilibrium of the ice sheet. It is not clear what this correction represents, how large it is, and how it evolves with time as the model evolves from its steady-state. Also, how can the dynamics of the model be compared to observations that show clear evolution trends, if the model is artificially forced to be close to a steady-state?

p.7 l.220-225: This part is not very clear. Also it does not seem very natural to force the velocity. If the velocity at the flux gates is prescribed there is not much freedom anymore in the model. It would make more sense in my opinion to force the ice front position for example.

p.8 l.249: What is the treatment of the calving front? What is the calving law used and how does the ice front evolve in the code? Why is the ice front allowed to retreat but not advance?

p.13 l.431-433: This sentence is not clear as it seems that all models start from the same thickness and same datasets in general. As the flux correction is applied to ensure that the model remains close to a steady-state, it is not surprising that the dynamic model and persistent representation remain close to each other.

p.14 l.465: The physical ice sheet models have not yet reach a state where they can be reliable and compare with observations and capture the dynamic processes at play. Some important physical processes could even be missing. So it might be unfortunately a little premature to pretend to compare observations with models, even if this remains a long-term objective. For these reasons, the distinction of more accurate models with this tool seems difficult to achieve, and results presented here are largely influenced by the flux correction applied in the physical models.

p.14 l.466: "dynamic models that account for known changes in ice dynamics": this statement is a little biased in my opinion, as the model is forced with observed velocities to reproduce the dynamic signal, while the physics alone should be able to reproduce this effect.

p.14 l.501: As mentioned above, it would be more natural to constrain the evolution of ice front position with observations instead of constraining the velocity. Ice front retreat triggers acceleration and would have a similar effect but would include the physical processes involved in this process.

[Figure]

p.15 l.518: It seems difficult to asses if sophisticated physical models perform better as their evolution is largely influenced by the flux correction. When such a correction is not applied, many models are not even capable of reproducing a mass loss for the Greenland ice sheet (Bindschadler et al., 2013).

**3  Technical comments**

Model names "SMB-only" and "RACMO-SMB-only" are rather confusing, this should be clarified.

p.5 l.138: What is $C\_20$?

p.7 l.213: How large is FC?

p.8 l.249: How is the ice margin retreating in CISM? This is not something commonly used and should be detailed. What is the criterion for calving? for moving the ice front?

p.9 l.281: Remove one "thickness"

p.10 l.298: How are the values averaged? What happens when cells are split between several GRACE-like grids?

p.11 l.355: add parenthesis after Figure 11

p.13 l.403-404: Rephrase

p.14 l.440-441: No clear: why do they only partly account for surface elevation changes? The RACMO data should include that. What is missing?

p.15 l.495: Missing parenthesis after (Joughin et al., 2004)

p.16 l.334: "in prep." → the paper appeared in GMDD

p.16 l.541: What about velocity observations? That would be a very valuable metric for

**[GMDD](javascript:void)**

dynamic changes.

p.18: It would be easier to have references listed alphabetically.

p.26 Fig.7: Rescale the yaxis

p.29 Fig.12: It might be easier to add letters on the subplots (instead of e.g. lower-right)

p.14 l.455: that they see HALF of the GRACE signal

**4  References**

Aschwanden, A., G. Adalgeirsdottir, and C. Khroulev, Hindcasting to measure ice sheet model sensitivity to inital states, The Cryosphere, 7, 1083–1093, doi:10.5194/ tc-7-1083-2013, 2013.

Aschwanden, A., M. Fahnestock, and M. Truffer, Complex Greenland outlet glacier flow captured, Nature Communication, 7, doi:10.1038/ncomms10524, 2016.

Bindschadler, R., et al., Ice-sheet model sensitivities to environmental forcing and their use in projecting future sea-level (the SeaRISE project), J. Glaciol., 59(214), 195– 224, doi:10.3189/2013JoG12J125, 2013.

Seroussi, H., M. Morlighem, E. Rignot, E. Larour, D. Aubry, H. Ben Dhia, and S. S. Kristensen, Ice flux divergence anomalies on 79 North Glacier, Greenland, Geophys. Res. Lett., 38, doi:10.1029/2011GL047338, 2011.

---

## Author Comment (AC1) · 27 Aug 2016

We discuss this point already with respect to our basin-only comparisons. By going to any finer resolution than that, we risk not having enough data (points) for the comparisons to be meaningful / interpretable. To some extent, this is an artifact of problems inherent to the ICESat campaign, and the much sparser spatial coverage and repeat coverage than was initially anticipated. As the reviewer suggests, and as we note in the conclusions, we expect this problem to become less of an issue as new data from other satellites come on line and are made available to the CmCt. Note that we have added a sentence to the 3 paragraph of section 6 to clarify this.

**Response to additional points**

*l210: "(the time series of spatially integrated, net SMB, relative to the 1960-1990 mean, is implied by the black-dotted line in Figure 4).": the time series in Figure 4 is really the integrated net SMB. I think the "implied" can be removed?*

The word "implied" has been changed to "indicated" in the revised manuscript.

*Bottom of p7, top of page 8: discussion on the dynamic forcing: not sure I fully understand the part "starting in 1999, we "play back" the converted time series...". I understand that there is no forcing from 1991 to 1999, then the forcing is applied as Dirichlet conditions relative to the 1999 model velocities? More precisions are needed for the choice of the location for the Dirichlet conditions. Why "several kilometres upstream from observed grounding line"? How is it chosen? A new figure made from Fig.1 with zooms on particular outlets glaciers could be useful.*

We have re-written this section of the paper to be clearer. It now reads:

"Under the assumption that the Enderlin et al. (2014) flux changes are dominated by velocity changes (discussed further below), we apply these changes in our model by first converting the data to a velocity increase *relative to* the 1999 observations (all 22 outlet glaciers would have a relative flux increase of 1 in 1999, i.e. no increase). Starting in 1999, we "play back" this time series, relative to the modeled 1999 velocity field, as the model is marched forward in time. In this way, the Enderlin et al. (2014) velocity changes are applied as Dirichlet boundary conditions on the model velocity several km upstream from observed grounding line locations (i.e., at the same locations as the Enderlin et al. (2014) observations)."

We have also added a zoom to the Figure 1, showing in more detail what a single flux gate looks like.

*Page 9: "Processing of model Output and observations". I understand that model output should be processed before online submission to the CMCT website. There is no documentation on the website (at least as far as we don't ask for login informations); It would be useful to give a table with the variables, their units, the time dimension, etc . . . that should be included in the netcdf for processing by CMCT.*

A users guide for the CmCt is in the process of being written and contains this (and other relevant) information. When that users guide is ready, it will be clearly linked to and made available from the CmCt website.

*L355: "e.g. Figure 11", missing closing ")"*

Corrected.

*Fig. 10: labels of the colorbars are not visible.*

We have added some text to the caption to indicate that the colorbars refer to the relative spatial weights applied when calculating the whole-ice-sheet mass changes (the color contoured values have no units).

*Fig. 11 caption : "kernal" → kernel*

Corrected.

*A paper comparing GRACE data with flow model simulations has just been published in The Cryosphere (Alexander et al., 2016). It could be cited and discussed.*

A reference to Alexander et al. (2016) has been added to the introduction where related previous work is mentioned. Also, we now mention additional similarities between simulations conducted by these two different efforts in our Results and Discussion section (section 6).

*References are not in alphabetical order; modify with respect to GMD requirements for references.*

Corrected (an incorrect style file was used in the submitted version of the manuscript).

---

## Author Comment (AC2) · 27 Aug 2016

Response to Anonymous Referee #2: "Interactive comment on *An ice sheet model validation framework for the Greenland ice sheet*"

We thank the reviewer for the detailed comments, each of which is addressed in our response below. For some comments, it was not entirely clear what the reviewer was requesting. We state where this is the case and note that we are happy to address any additional concerns once we better understand what those concerns are.

**Response to General Comments**

*The currently included and in the future planned observations are based on satellite products, which biases the observation-based validation towards surface properties, such as elevation and its change. What's about ice sheet-internal properties such as temperature profiles from existing ice cores or radar detected layers, that could help to verify the internal structure, which may be important for the susceptibility of the ice sheet to applied forcing in the future?*

We agree with the reviewer that remote sensing observations reflect a bias for the ice sheet surface, and that exploiting internal information (e.g., layer structure, such as that of MacGregor et al., 2015) is a logical next step. At present however, we have chosen to first exploit our core strengths, which are surface-based remote sensing data from space. While exploiting the wider suite of geophysical observations remains a long-term goal of the CmCt, in the present paper we have chosen to focus only on the current and near-term capabilities.

**Response to Specific Comments**

*L52: Could you please add after web-address that the service is available after registration.*

In order to not clutter the paper with minor details, we'd prefer to leave this information out. Once going to the link, it is made clear that one must register for an account before using the service. We are confident that the registration requirement will not deter potential users.

*L90-21: I'm sorry, but I haven't understood without a doubt what is meant. Please clarify.*

Unfortunately, it is not clear what lines the reviewer is referring to here. We are happy to clarify once we found out what the problem is.

*L101-107: You may add a quantity such as "number of good data points" to guide the user to identify "good" years as the mentioned years 2003, 2004 and 2007.*

It's not clear to me what the reviewer is asking for here. In the lines pointed to, we make

it clear what factors led us to choose observations from years 2003, 2004, and 2007. Note that the data files returned to the user from the CmCt include information on the number of observational ("good") data points used in the model-to-observation comparison.

*L141: Since some may think instead of the calendar year (starting 01.Jan) about the hydrological year (starting 01.Sept). Therefore, could you please clarify.*

We have changed the preceding sentence as follows: "The monthly data are annually-averaged (based on the calendar year), …"

*L151: Interpolation are bi-linear, conservative, ...?*

We've clarified that bilinear interpolation was used when creating the initial condition data sets.

*L154: Interpolation are bi-linear, conservative, ...?*

We've clarified that SMB data were interpolated onto our ice sheet grid using barycentric interpolation (this was necessary because the RACMO2 grid is not perfectly regular).

*L166: Do you mean:"... and another that evolves its surface elevation only according to ... ."?*

Yes. We have updated the text in this sentence to be clearer.

*L206: Do you mean:"... and optimization of sliding parameters to meet observed velocities, the ... ."?*

Yes. This sentence has been updated as suggested.

*L229-234: Ice beyond the flux gates are not taken into account for the comparison. Does it still exists and interacts with the ice flow upstream?*

Yes. As written, we think it is clear that the removal of ice downstream of the flux gates is done as a post-processing step, thus implying that it does not affect the model during runtime.

*L243: Do you mean:"... changes in ice thickness are always an order of magnitude ... ."?*

Yes. We've added the word "always".

*L250: The application of the mask to remove ice beyond the initial footprint acts as a sink. How large is this sink term? How large is its factional size compared to the applied total SMB.*

In general, relative to the total (area integrated) SMB during any time step this term is negligibly small. As noted in the paper, this procedure is largely in place to deal with the limited footprint of existing SMB data, which would lead to unchecked growth and

advance for any ice that advances beyond the (fixed) coverage of the RACMO2 SMB data. This is discussed in detail in the $2^{nd}$ to last paragraph of section 3.2.

*L252-254: You may split the sentence in two pieces:"... advance in limited marginal regions. In reality, strong negative mass balance at ... ."*

This sentence is grammatically correct as written, in which case we prefer to leave it as is.

*L261: Have I understood you correctly:"... steps, RACMO2 SMB fields are applied to either increase (positive SMB) or decrease (negative SMB) the ice thickness accordingly." ?*

Yes. We think that the positive or negative action of applying the surface mass balance field is implicit in the sentence as currently written.

*L276-277: How large are the 2% compared to the other presented mass changes?*

It's unclear what "other presented mass changes" are being referred to by the reviewer. The point is simply that 0.2% of the ice sheet area present in the initial condition is not used when comparing the model output to observations (this area of the ice sheet is not included in either the model or observational data). So this number does not really represent a mass change in any physical sense of the word. Also, note that the number presented in the paper is **0.2%** rather than **2%**.

*L281-284: Are corner points needed for each grid box. If optional, would the tool box than perform conservative interpolation?*

Corner points are not needed by the CmCt. Also, while not discussed, we have tried a number of different interpolation schemes at the stage discussed in this part of the paper (bilinear, nearest neighbor, conservative) and found the results to be either trivially changed or not at all changed.

*L298: I guess you mean:"... averaged weighted by grid size."?*

We don't actually mean that here, because at present, the CmCt is only set up to work with uniform resolution grids. Eventually, when the tool is altered to allow for the use of variable resolution grids, we would need to apply some weighting based on grid cell size.

*L299-301: Do you mean:"... cells by the ratio of the ice-covered model grid area in that cell to the total cell area... ."?*

Yes. We've made that clarification in the revision.

*L302: Is it possible to change the density of the ice on the web page?*

Currently, the ice density is not a configurable parameter in the GRACE processing module. As more functionality is added, we will consider making the ice density a

configurable parameter.

*L355: You may dropped the closing bracket:"... of time (e.g. Figure 11), and ... ."?*

Corrected.

*L380: You mean probably mean:" ... of at least 15 cm of water over the period 2003-2012 ... ."?*

Yes. We have added this clarification in the revised text.

*L385: You may change:"... GRACE observations has a PVE=100 at each location ... ."?*

We've changed the text to the following: "A model which perfectly reproduces the GRACE observations has a PVE=100 in each 0.5 x 0.5 degree bin."

*L388: Since math (equation 4) is clear you may change:" ... the observed variance exhibit negative PVE values"*

It's not exactly clear where the reviewer is suggesting a change be made. Further, we think the sentence being pointed to adequately explains how a negative PVE value should be interpreted.

*L391: You may replace $M_{Trend}$ with $\delta M_{Trend}$ to highlight that it is a difference between two trends.*

As this is merely a matter of preference, we prefer to leave the definition as is. The "M" is in reference to "metric", and the subscript "trend" clearly indicates that this metric is with respect to the mass trends.

*L426: You may write:" ... may be too small that they ... ."?*

We think this sentence is clear enough in meaning as written (and "… may be too small that they …" does not appear to be grammatically correct).

*L436: You may emphasize:"... hat initially good match (Figure 8) on a period covering a decade."*

It's not clear that the suggested revision improves the clarity of the existing sentence, so we prefer to leave it as is.

*L510: This, of course, only applies if the driving model climate is identical or at least very similar to the current climate. Otherwise I expect that coupled climate-ice sheet models would lead to an ice sheet geometry that differs from the observed state.*

We agree. What we meant here by "more meaningful" was that the examples discussed in the paper were mainly meant for illustration purposes, but that the tool could be far more useful if the goal was to assess performance of a model receiving its forcing through

coupling to an ESM (no changes were made to the manuscript).

*L569-694: Reference. What's the style and sorting criteria for Geoscientific Model Development? If none, please sort using the last name and also have a consistent way to write the first names.*

Corrected (an incorrect style file was used in the submitted version of the manuscript).

**Response to comments on Tables**

*Table 1 and 2: Please give a remark that "Pers" means persistence and you may also indicate "CISM" as Community Ice Sheet Model and RACMO2 as regional model.*

Both tables now contain information to this effect in their captions.

**Response to comments on Figures**

*Figure 1: Could you add to the color bar of the first two left sub-figures (observed and modeled velocities) a second row that lists instead of log10(v) the actual velocity (v)?*

In order to avoid cluttering the figure further, we would prefer not to add an additional colorbar (note that the colorbar showing the difference between the model and observations is in linear units).

*Figure 1: In the right figure (velocity difference) the black dots representing the flux gates and the negative velocity differences (blue color) are hard to distinguish. What's about adding a tick ("-") slight away from the ice margin to indicate the locations of the flux gates?*

In response to reviewer no. 1, we have added a zoom to the figure, showing in more detail what a single flux gate looks like. We prefer not to clutter this already busy figure with too much additional information.

*Figure 3: To avoid any ambiguity you may change the caption slightly:"... from greatest (Jakobshavn) to least (Nordre) flux in 1999."*

The caption has been corrected as suggested.

*Figure 4: Could you increase the line thickness please?*

Line thickness has been increased.

*Figure 7: Since all dots seem to fall within -300 and 300, you may reduce the ordinate axis to this range.*

The extent on the vertical axis of this figure has been changed to +/- 500 m.

*Figure 8: What do you think about increasing the size of the black entire ice sheet symbols?*

We prefer to leave the symbols the same size, as it might not be clear why one set of symbols was larger than the other.

*Figure 9: Since the figure caption indicates a mass change the unit shall be accordingly. You may write:"... Units are meters of water equivalent height / 10 years." or "... Units are meters of water equivalent height for the given period."*

The caption is correct as written. The mass change shown is the *total mass change* (as written) occurring from 2003-2012.

*Figure 11: You may change the caption:"Observed and modeled, cumulative whole-ice sheet trends ... "?*

Changed as suggested.

*Figure 12: To guide the read just spotting the figures what is meant, you may add:" ... percent of GRACE variance (Eqn 4) explained by ... ."?*

Changed as suggested.

**References**

MacGregor, J. A., and Coauthors, 2015: Radiostratigraphy and age structure of the Greenland Ice Sheet. *J Geophys Res-Earth*, **120**, 212–241, doi:10.1002/2014JF003215.

---

## Author Comment (AC3) · 27 Aug 2016

**Response to General Comments**

*My main concern is that the "most sophisticated" dynamic model reproduces the trends in altimetry and gravity for the wrong reasons. Correctly initializing ice flow models remains a challenge today and an active area of research (Seroussi et al., 2011; Aschwanden et al., 2013; 2016) due to the lack of reliable observations and the long response time of ice sheets. In order to bypass this problem, the authors impose an additional steady-state surface mass balance (SMB) on top of the climate derived SMB, therefore forcing the model to remain close to its initial conditions. Without this unphysical forcing, the model would likely diverge quickly from this initial state.*

We appreciate and agree with these criticisms and acknowledge them in our paper. In our final few concluding sentences, we note that an important and necessary improvement is model initialization techniques that would allow for the use of non-anomaly-based forcing and / or that would allow for initialization with realistic transients (note that some of this work has been (and continues to be) done by authors on this paper, e.g., Perego et al. (2014)). In our revisions, we have expanded on this discussion to further emphasize its importance.

We also note that the application of flux corrections and anomaly forcing is not unique to this study and, at the beginning of section 3, we've noted several other recent publications that use these same techniques in studies of Greenland ice sheet dynamics. Most importantly, we emphasize that the main point of this paper is not to present and validate a perfect ice sheet model simulation (or to claim that we are validating a particular ice sheet model), but rather to demonstrate that the validation tool and metrics proposed here are useful for discerning between simulations. We have added a final paragraph to the introduction to explain this and make sure our intent is clear up front.

*Furthermore, the physical model has to be not only forced with an unphysical SMB, but also constrained with imposed velocities over the most dynamic areas, in order to get close enough to the observed changes in gravity. At this point, there is not much physics left in this model.*

We acknowledge these same shortcomings in 3$^{rd}$ paragraph of the Conclusions. At the

same time, we disagree with the suggestion that "there is not much physics [=dynamics?] left in this model." The response of an ice sheet model to an applied forcing will vary immensely depending on the physical fidelity of that model. For example, we certainly do not expect that a model based on the shallow-ice approximation would be able to simulate the inland propagation of acceleration and thinning from perturbations at its margin as accurately as a model based on a first-order approximation to the Stokes equations (due to the lack of horizontal stress gradients). This expectation applies regardless of if those perturbations are applied in the somewhat artificial way used in this paper, or if those perturbations are applied through much more realistic model physics (e.g., a physically-based calving law and an evolving ice front) that serve as an intermediary between the model and external climate forcing (e.g., ocean warming and ice tongue melting). In this sense, we agree that the model is lacking "physics" (i.e., important physical processes and parameterizations) that one might like to test (as are most models), but the dynamic response of the model to an applied forcing is still worth testing. Here, it is that realistic dynamic response (first to only SMB forcing and then to SMB and outlet glacier dynamics forcing) that leads to (1) differences between our ideal vs. numerical model simulations, and (2) differences between numerical model simulations using different forcing (and as we show, these differences can be detected and quantified using the CmCt and our proposed metrics).

*So if I don't question the development of this framework or the benefits to have a tool that can easily compare observations and model results, I am wondering if any validation of numerical models can be done today given the many improvements still needed by these models to become more accurate. I would therefore recommend to reduce the emphasis of this framework as a validation tool and rewrite the manuscript accordingly.*

As noted above (and pointed to in the paper), there are currently significant advances being made in initialization methods, which result in realistic model steady states that are in agreement with observations AND that are in approximate equilibrium with specified climate forcing fields. And while we agree that many other improvements are still needed before models are accurate enough to make full use of such a validation tool, we would argue that those model developments and the development of tools, methods, and metrics for validation should happen in parallel. Indeed one can imagine model deficiencies that are presently unknown and that cannot be identified or remedied without the tools to aid in model validation.

Because many of the same concerns noted above are also repeated below, we address them further on a point-by-point basis there.

**Response to Specific Comments**

*In the abstract (l.6) as well as several other places in the manuscript, the author qualify the static representation and the SMB only applied to this static representation as "non-dynamic models", which is misleading, as these representations do not include any physical model. The text should better distinguish between physical models and other*

*representations.*

The use of the word "model" refers, in this case, to a conceptual model, and we think that the qualifier "non-dynamic" is clear. Nevertheless, we have edited the section of the abstract the reviewer notes. The relevant sentence has been changed to:

"Based on basin- and whole-ice-sheet scale metrics, we find that simulations using both idealized conceptual models and dynamic, numerical models provide an equally reasonable representation of the ice sheet surface (mean elevation differences of <1 m)."

We also note that in two sections of the text, we do already go into detail about what these idealized models are based on (the 2nd paragraph of the section 3 introduction (lines 165-170 of original submission) and the last paragraph of section 3.2 (lines 258-265 of original submission)).

*p.1 l.17: "simulations of varying complexity": there is no varying complexity found in the models presented here (same stress balance approximation, ...). What varies between these simulations is the degree of "forcing" of the models, the velocity being applied as a Dirichlet forcing for one of the simulations.*

We specify "**simulations** of varying complexity" as opposed to "**models** of varying complexity" for the exact reason the reviewer points out. Further, some of our simulations are based on idealized or conceptual models (arguably a simulation conducted using a dynamic, numerical model is more complex than a simulation based on an idealized conceptual model). We believe that the current descriptions for how the simulations differ from one another (section 3.2) is adequate to guide the reader in understanding what we mean by "varying complexity".

*p.2 l.20-21: The "most sophisticated" models are actually forced by imposing the velocity to be equal to observations at the border of the domain. Showing that such a simulation exhibits mass trends similar to gravity observations does not prove any predictive capability of the model. First, the velocity is not produced by the model but forced in all the dynamic areas. Furthermore, the flux correction applied to the physical models introduces a large mass change that prevents to compare simulations to observations. What this result mainly shows is that observations are rather consistent and that a large part of the mass change signal can be explained by changes in SMB and acceleration.*

Again, proving predictive capability of the model is not the goal of this effort, as we clearly state at multiple points of the paper. The goal, as stated, is to demonstrate that the framework can distinguish between simulations of differing realism with respect to observations. That a simulation forced by the most complete set of forcing data exhibits similar mass trends to the observations is precisely the point; we expect that such a simulation will "score" much better than (i) a simulation forced by an less complete set of data (e.g., our "SMB-only" simulation) and / or (ii) simulations that do not allow for any dynamic response of the ice sheet (our "persistence" and "RACMO2-SMB-only" simultions). Recall that our stated goal is to demonstrate that our validation tool and metrics can distinguish between simulations with different levels or realism with respect

to the observations. In order to demonstrate that, we first need to create simulation output that we expect will have those characteristics.

*p.2 l.23: What about velocity changes? I would imagine that comparing the dynamic signal would be an important part of such a tool, as the main objective of an ice sheet model is to reproduce the dynamic signal, not the SMB.*

Here, we examine the dynamic signal through thickness and mass changes. In our opinion, there are not yet enough velocity data with wide enough coverage to make those data useful for large-scale model validation. Also, at present, many models (including the one discussed here) use the available velocity data in their initialization procedure. As more velocity data become available in the future, they could certainly be included. For now, we've chosen to focus on data that is a proxy for dynamic evolution via changes in ice sheet thickness (or mass).

*p.2 l.33-37: I don't agree with this statement. The main problem is that ice flow models are still unable to reproduce the observed changes, and that improving their initial conditions is a very active area of research. Models therefore have to either run spin- up or apply flux correction (similar to what is done in this manuscript), to get closer to observations. Accurately comparing model results and observations is therefore beyond the scope of most studies since lots of models cannot even capture the correct trend (Bindschadler et al., 2012).*

As we've noted above, we do discuss and acknowledge problems, and future improvements, related to model initialization. And we would argue that validation tools should be developed in parallel to models, so that both can be improved simultaneously (rather than sequentially).

*p.7 l.208: The authors explain how they apply an SMB correction to get closer to a steady-state equilibrium of the ice sheet. It is not clear what this correction represents, how large it is, and how it evolves with time as the model evolves from its steady-state.*

It is clear from Equation 1 (and the related discussion) that the flux correction is a correction to the RACMO2 SMB forcing. We have added the word "static" to its initial description to clarify that it is unchanging in time (this should also be clear from Equation 1). We have also added references to several other recent papers where similar methods have been applied to SMB forcing in studies of Greenland ice sheet dynamics (Price et al., 2011; Shannon et al., 2013; Nowiki et al., 2013; Edwards et al., 2014). Also, we have added discussion to this same section noting that this process assumes that in 1990 Greenland was in equilibrium with its mean SMB over the previous three decades. Thus, perturbations, and volume trends following from them, are relative to this assumption of steady-state in 1990.

*Also, how can the dynamics of the model be compared to observations that show clear evolution trends, if the model is artificially forced to be close to a steady-state?*

The model is forced to be in steady-state in 1990 based on the longer-term mean SMB. Thus, modeled volume evolution trends will be a result of forcing applied after 1990.

Anticipating that this will result in a reasonable comparison between model and observations follows from the assumption that the bulk of Greenland's volume / mass change over the past few decades is a result of forcing applied over that time period (as opposed to resulting from some longer term transient). In turn, this assumption follows from satellite-based reconstructions of Greenland's net mass balance, which suggest that it was in quasi-equilibrium during the 1980's and 1990's (van den Broeke et al., 2009; Fig. 2), at least relative to *after* the mid-1990's and through to the present day.

We appreciate that the reviewer does not favor the use of anomaly SMB forcing, and we do acknowledge this as a shortcoming in our paper. Nevertheless, Figures 4 and 11 do show that modeled mass trends following from our applied SMB forcing provide a reasonable approximation of the observations in the sense expected they would (volume changes resulting from SMB-only forcing underestimate the overall mass loss to a greater degree than simulations that also account for mass loss via outlet glacier flux changes).

*p.7 l.220-225: This part is not very clear. Also it does not seem very natural to force the velocity. If the velocity at the flux gates is prescribed there is not much freedom anymore in the model. It would make more sense in my opinion to force the ice front position for example.*

We agree it would be more desirable to force the ice front position. But doing so in a manner that was able to match the actual flux observations is something that is beyond possible right now for any model (as far as we know). As noted above (and as discussed in the paper), the goal here is not to do a perfect job on the model simulations, but to show that simulations that we expect to do relatively better or worse with respect to matching the observations can be shown to do so using our tool and the validation metrics we propose (we also note that this section has been edited as per comments from other reviewers).

*p.8 l.249: What is the treatment of the calving front? What is the calving law used and how does the ice front evolve in the code? Why is the ice front allowed to retreat but not advance?*

There is no calving law applied. The ice front position is assumed to stay fixed in time and space and any ice that reaches it is removed from the domain. Reasons for why the ice front is allowed to retreat but not advance (and arguments for why this is a reasonable simplification) are discussed on lines 246-256 (original version). Again, we note that (1) these same simplifying approximations have been made in numerous other modeling-based studies of Greenland ice sheet dynamics (see references listed above and below) and (2) the purpose of this study is not to provide a perfect model simulation or argue that we have validated a particular model. The purpose is to provide model outputs that are quasi-realistic with respect to observations to demonstrate the use of the validation tool and metrics. We have added some text to this section of the paper to make it clear that no explicit calving model is used.

*p.13 l.431-433: This sentence is not clear as it seems that all models start from the same thickness and same datasets in general. As the flux correction is applied to ensure that*

*the model remains close to a steady-state, it is not surprising that the dynamic model and persistent representation remain close to each other.*

The point of these lines (and which we think is clear as written) is that the model initial condition is based on a DEM that was constructed using some of the same ICESat observations that we are comparing our model output against. In this sense, the model initial condition is already somewhat biased to "look" like the ICESat observations.

*p.14 l.465: The physical ice sheet models have not yet reach a state where they can be reliable and compare with observations and capture the dynamic processes at play. Some important physical processes could even be missing. So it might be unfortunately a little premature to pretend to compare observations with models, even if this remains a long-term objective. For these reasons, the distinction of more accurate models with this tool seems difficult to achieve, and results presented here are largely influenced by the flux correction applied in the physical models.*

This sounds like a value judgment and one that we disagree with. In the paper, we acknowledge shortcomings in model initialization methods, model physics, and simplifications made in the modeling conducted here. But we would argue that there is no reason to wait for a perfect model before assembling the tools to be used for validation models and making those tools as widely available and useful as possible. In our opinion, model development and the development and testing of validation tools, should be conducted in parallel.

*p.14 l.466: "dynamic models that account for known changes in ice dynamics": this statement is a little biased in my opinion, as the model is forced with observed velocities to reproduce the dynamic signal, while the physics alone should be able to reproduce this effect.*

Our statement says nothing about whether or not we include appropriate model physics (and nowhere do we claim to have accounted for all necessary model physics). Our claim here is that model dynamics, when forced appropriately, are capable of mimicking observed dynamic changes, and that when ignoring some forcings or using a model that does not respond dynamically, the ability of a model to reproduce observed dynamic changes is measurably degraded.

We further note that a model could have a perfect and complete representation of dynamics and physics (e.g., calving, subglacial hydrology), and without applying the correct forcing (e.g., SMB or submarine melt rates), that model would still fail to match observations. Here, we are showing that IF models could translate the appropriate climate forcing to the correct boundary perturbations (e.g., IF the correct SMB and submarine melt rates were applied by a climate model, and IF the coupling and model physics then translated those correctly to land ice model perturbations), then models can be expected to provide a realistic response, relative to observations.

*p.14 l.501: As mentioned above, it would be more natural to constrain the evolution of ice front position with observations instead of constraining the velocity. Ice front retreat*

*triggers acceleration and would have a similar effect but would include the physical processes involved in this process.*

While ice velocity / flux time series with relatively complete spatial and temporal coverage do exist, we are not aware of any similarly detailed and complete datasets for ice front position. Further, if such a dataset did exist, it is not clear how it would / could be used to force a large-scale model. Would a time series of calving rates need to be specified for each outlet glacier, in order to force the appropriate ice front motion? Again, such an effort, if feasible, would be well beyond the stated goals of the modeling in this paper, which is to produce realistic outputs that we can use to demonstrate the use of our validation tools and metrics. As noted above, we have now made this goal very explicit at the beginning of the paper.

*p.15 l.518: It seems difficult to asses if sophisticated physical models perform better as their evolution is largely influenced by the flux correction. When such a correction is not applied, many models are not even capable of reproducing a mass loss for the Greenland ice sheet (Bindschadler et al., 2013).*

Our numerical ice sheet model maintains a steady-state when forced by the mean, 1960-1990 SMB with a flux correction applied. When this same flux correction is applied to the time series of RACMO2 surface mass balance and that anomaly time series is used to force our ice sheet model, the result is a change in ice sheet mass over the 2002-2012 time period that is ~60% of the observed change (red line in Figure 11 – recall that we don't expect the SMB forcing alone to be capable of recreating the full fraction of the observed change). This speaks to the effectiveness of forcing a dynamic ice sheet model with anomaly SMB forcing. Further, if we compare the *difference* between our model simulation forced only by SMB anomalies and the observations, we find that both the sign and the magnitude of the mass difference (+700 Gt in 2012) are very similar to the *difference* found for similarly targeted simulations in other studies that do *not* use anomaly SMB forcing methods (In Alexander et al. (2016), the difference between the simulated and observed cumulative mass anomaly in 2012 relative to 2003 – the same observational time period considered here – is also approximately +700 Gt). We have added a sentence pointing out this level of similarity in the second to last paragraph of section 6.

**Technical Comments**

*Model names "SMB-only" and "RACMO-SMB-only" are rather confusing, this should be clarified.*

In 3.2 of the paper, we clearly state what these two simulations consist of and how they differ from one another.

*p.5 l.138: What is C_20?*

C_20 is a harmonic term. We've updated the text so this is now clear.

*p.7 l.213: How large is FC?*

For 2003 to 2009, the time-averaged, spatially integrated SMB for Greenland, based on our RACMO2 forcing dataset, is approximately 330 Gt/yr (in agreement with values reported by van Angelen et al. (2013), Table 2). The spatially integrated value of our flux correction (FC), is 297 Gt/yr (~10% smaller).

*p.8 l.249: How is the ice margin retreating in CISM? This is not something commonly used and should be detailed. What is the criterion for calving? for moving the ice front?*

See response to related discussion above.

*p.9 l.281: Remove one "thickness"*

Corrected.

*p.10 l.298: How are the values averaged? What happens when cells are split between several GRACE-like grids?*

Ice thickness values on the 1 km ice sheet grid are treated as point masses. As such, they are assumed to "belong" to whichever 0.5 deg. grid cell they fall into. Given the disparity in size between a 1x1 km ice model grid cell and a 0.5x0.5 deg. GRACE grid cell, this is a reasonable simplification.

*p.11 l.355: add parenthesis after Figure 11 p.13 l.403-404: Rephrase*

Corrected.

*p.14 l.440-441: No clear: why do they only partly account for surface elevation changes? The RACMO data should include that. What is missing?*

The RACMO forcing data are in ice equivalent units. For a given mass change, the relative thickness change for solid ice will be approximately half that for the same mass of firn (because near-surface firn has a density approximately half that of solid ice). Thus, seasonal elevation changes in the model due to the application of (ice equivalent) SMB forcing will be a muted expression of the seasonal elevation changes seen by ICESat. We have added some clarifying text to this paragraph of the manuscript.

*p.15 l.495: Missing parenthesis after (Joughin et al., 2004)*

Corrected.

*p.16 l.334: "in prep." → the paper appeared in GMDD*

Corrected.

*p.16 l.541: What about velocity observations? That would be a very valuable metric for dynamic changes.*

Please see related discussion above.

*p.18: It would be easier to have references listed alphabetically.*

Corrected (an incorrect style file was used in the submitted version of the manuscript).

*p.26 Fig.7: Rescale the yaxis*

This was also suggested by another reviewer and has been corrected.

*p.29 Fig.12: It might be easier to add letters on the subplots (instead of e.g. lower-right)*

Corrected for Figures 9, 10, and 12.

*p.14 l.455: that they see HALF of the GRACE signal*

Corrected.

**References**

Alexander, P. M., M. Tedesco, N.-J. Schlegel, S. B. Luthcke, X. Fettweis, and E. Larour, 2016: Greenland Ice Sheet seasonal and spatial mass variability from model simulations and GRACE (2003–2012). *The Cryosphere*, **10**, 1259–1277.

Perego, M., S. Price, and G. Stadler, 2014: Optimal initial conditions for coupling ice sheet models to Earth system models. *J. Geophys. Res. Earth Surf.*, **119**, doi:10.1002/(ISSN)2169-9011.

van den Broeke, M., J. Bamber, J. Ettema, and E. Rignot, 2009: Partitioning recent greenland mass loss. *Science*, **326**, 984–986, doi:10.1126/science.1178176.

van Angelen, J. H., M. R. Van Den Broeke, B. Wouters, and J. T. M. Lenaerts, 2013: Contemporary (1960–2012) Evolution of the Climate and Surface Mass Balance of the Greenland Ice Sheet. *Surv Geophys*, **35**, 1155–1174, doi:10.1007/s10712-013-9261-z.

Velicogna, I., and J. Wahr, 2013: Time-variable gravity observations of ice sheet mass balance: Precision and limitations of the GRACE satellite data. *Geophys. Res. Lett*, **40**, doi:10.1002/grl.50527.

---

## Author Response (AR1)

Point-by-Point response to reviews for, "Interactive comment on *An ice sheet model validation framework for the Greenland ice sheet*"

We note that point-by-point responses to reviewer comments are provided in the individual responses to each reviewer.